# Development of visual response selectivity in cortical GABAergic interneurons

Jeremy T. Chang [ID] [1 ✉] & David Fitzpatrick [ID] [1]

The visual cortex of carnivores and primates displays a high degree of modular network organization characterized by local clustering and structured long-range correlations of activity and functional properties. Excitatory networks display modular organization before the onset of sensory experience, but the developmental timeline for modular networks of GABAergic interneurons remains under-explored. Using in vivo calcium imaging of the ferret visual cortex, we find evidence that before visual experience, interneurons display weak orientation tuning and widespread, correlated activity in response to visual stimuli. Robust modular organization and orientation tuning are evident with as little as one week of visual experience. Furthermore, we find that the maturation of orientation tuning requires visual experience, while the reduction in widespread, correlated network activity does not. Thus, the maturation of inhibitory cortical networks occurs in a delayed, parallel process relative to excitatory neurons.

[1] Department of Functional Architecture and Development of Cerebral Cortex, Max Planck Florida Institute for Neuroscience, Jupiter, FL, USA.
✉email: jeremy.chang@mpfi.org

The onset of visual experience coincides with a period during which cortical circuits are highly plastic. This enables visual experience to drive the maturation of functional properties and network organization of excitatory neurons, improving orientation and direction selectivity and establishing binocular alignment of response preferences[1–5]. The visual cortex of carnivores and primates has a modular arrangement of excitatory neurons, which is characterized by the local clustering and long-range patterning of responses and functional properties. Modular organization is evident in excitatory networks before visual experience, suggesting it develops in an experience-independent manner[4,6,7]. Working in concert with excitatory neurons are inhibitory, GABAergic interneurons (GABA-INs). In the mature ferret, GABA-INs are orientation-selective and coupled to the excitatory network—exhibiting a modular structure, where neighboring neurons share common orientation preferences[8]. The development of these aligned networks is a remarkable feat, as inhibitory and excitatory neurons derive from different progenitors and undergo distinct patterns of migration[9]. Furthermore, the interaction of inhibition and excitation is important for normal cortical function[10,11], and manipulations of inhibition alter the structure of sensory maps[12].

Despite the contribution of GABA-INs in cortical network development, the sequence of events that underlie the emergence of mature interneuron functional properties remains poorly understood. Developmental changes in the functional properties of mouse GABA-INs (becoming more broadly tuned with experience[13,14]) have been observed. However, it is unclear how studies in mice generalize, because they lack the modular structure and selective responses for GABA-INs found in higher mammals[15,16]. Here we examine the contributions of experience in the maturation of the functional properties and network organization of GABA-INs in the modular ferret visual cortex using two-photon and widefield epifluorescence calcium imaging.

We find that before eye opening, when excitatory neurons display orientation selectivity[4], GABA-INs are poorly orientation tuned and display variable responses. The immaturity of GABA-IN responses is also evident at the network scale where visual stimuli drive widespread patterns lacking the modular structure. These immaturities resolve shortly after eye opening, and evidence from deprivation experiments suggests that both experience-dependent and independent mechanisms contribute to this process. Thus, the development of functional properties and modular network structure of GABA-INs in visual cortex appears delayed relative to excitatory neurons, a chronology that suggests an instructive role for excitatory networks in the development of inhibitory circuits.

## Results

### GABA-INs develop orientation selectivity after eye opening.
Using viral expression of GCaMP6s under the mDlx enhancer, which preferentially labels GABA-INs[8,17], we assessed single neuron responses in layer 2/3 of ferret visual cortex to the binocular presentation of square drifting gratings with two-photon calcium imaging. We compared three groups of animals with different amounts of visual experience: those with no visual experience (at or before eye opening, "Naive"); those with a short period of visual experience (4 to 7 days of visual experience, "Brief"); and those with an extended period of visual experience (greater than 8 days of visual experience, "Extended"). GABA-INs in Brief and Extended experience animals exhibited strong orientation-selective responses that were not significantly different. (Fig. 1d–f, OSI Brief $0.54 \pm 0.06$ vs. Extended $0.52 \pm 0.05$, $p = 0.8298$, $n = 6$ vs. 5, Mean $\pm$ SEM, bootstrap significance test). Modest improvements were observed in the direction selectivity

(Fig. 1g, DSI Brief $0.16 \pm 0.03$ vs. Extended $0.24 \pm 0.03$, $p = 0.0444$, $n = 6$ vs. 5, Mean $\pm$ SEM, bootstrap significance test). Also, the fraction of significantly orientation tuned cells did not change (Fig. 1j, Brief $0.81 \pm 0.03$ vs. Extended $0.64 \pm 0.05$, $p = 0.0521$, $n = 6$ vs. 5, Mean $\pm$ SEM, bootstrap test), and the fraction of cells that were significantly direction tuned only modestly increased (Fraction of cells significantly direction tuned, Brief $0.17 \pm 0.02$ vs. Extended $0.28 \pm 0.03$, $p = 0.0214$; $n = 6$ vs. 5, Mean $\pm$ SEM, bootstrap test). Together, these data suggest that the orientation and direction selectivity of GABA-INs is near mature levels within 1 week of visual experience.

In contrast, GABA-INs in visually naive animals (Fig. 1c, Naive) exhibit significantly weaker orientation selectivity compared to experienced animals, with a mean orientation selectivity that is significantly lower (Fig. 1e, Naive $0.22 \pm 0.04$ vs. Brief $0.54 \pm 0.06$, $p = 0.0032$, $n = 7$ vs. 6; Naive $0.22 \pm 0.04$ vs. Extended $0.52 \pm 0.05$, $p = 0.0028$, $n = 7$ vs. 5; Mean $\pm$ SEM, bootstrap significance test). Naive animals also exhibited a lower percentage of orientation tuned cells (Fig. 1i, Naive $0.24 \pm 0.09$ vs. Brief $0.81 \pm 0.03$, $p = 0.0344$, $n = 7$ vs. 6; Naive $0.24 \pm 0.09$ vs. Extended $0.64 \pm 0.05$, $p = 0.0164$, $n = 7$ vs. 5; Mean $\pm$ SEM, bootstrap significance test). A small fraction of GABA-INs displayed direction selectivity; however, the overall proportion did not significantly change over the first week of experience (Fig. 1g, Naive $0.10 \pm 0.03$ vs. Brief $0.17 \pm 0.02$, $p = 0.0638$, $n = 7$ vs. 6, Mean $\pm$ SEM, bootstrap significance test), and selectivity modestly increased over the second week (Naive $0.10 \pm 0.03$ vs. Extended $0.28 \pm 0.03$, $p = 0.0040$, $n = 7$ vs. 5, Mean $\pm$ SEM, bootstrap significance test). One characteristic of visual responses in Naive animals that could contribute to the poor selectivity is trial-to-trial variability in the cellular responses- a phenomenon not seen in experienced animals (Fig. 1c–e). To quantify this, we calculated a variability index (see Methods). Naive animals show significantly higher variability than experienced animals (Fig. 1h, Naive $0.79 \pm 0.02$ vs. Brief $0.48 \pm 0.04$, $p = 0.0004$, $n = 7$ vs. 6; Naive $0.79 \pm 0.02$ vs. Extended $0.48 \pm 0.02$, $p = 0.0004$, $n = 7$ vs. 5; Mean $\pm$ SEM, bootstrap significance test). These findings suggest that before eye opening, GABA-INs are poorly orientation tuned in part because of cellular response variability and that changes in the week after eye opening led to reductions in variability and mature levels of orientation selectivity.

Does weak orientation tuning arise from visually-evoked activity? Were lack of visual responsiveness or spontaneous activity to play a role in poor orientation tuning, we would expect to observe fewer visually responsive neurons early in development. We do not observe this. We presented blank trials (trials without a stimulus) and characterized the proportion of GABA-INs that had responses to visual stimuli that were larger than two standard deviations as compared to blank trials. Most GABA-INs were visually responsive, and this fraction did not significantly change (Fig. 1i, Fraction of responsive cells, Naive $0.998 \pm 0.001$ vs. Brief $0.995 \pm 0.003$, $p = 0.3412$, $n = 7$ vs. 6; Naive $0.998 \pm 0.001$ vs. Extended $0.975 \pm 0.013$, $p = 0.0647$, $n = 7$ vs. 5; Mean $\pm$ SEM, bootstrap test). Therefore, the activity of GABA-INs observed during stimulus presentation reflects visually driven activity for all age groups.

While peak cellular responses reflect visually-evoked activity, one question that may arise is whether activity for the non-preferred orientation also reflected visually-evoked activity. To test this, we computed a Stimulus Activity Index (SAI, see Methods) by comparing responses to the non-preferred orientation and blank trials for cells that were significantly orientation selective. The SAI reflects the degree to which observed responses deviate from the predicted response arising from spontaneous activity. Positive SAIs reflect visually driven activity, negative SAIs reflect visually suppressed activity, and an SAI of zero

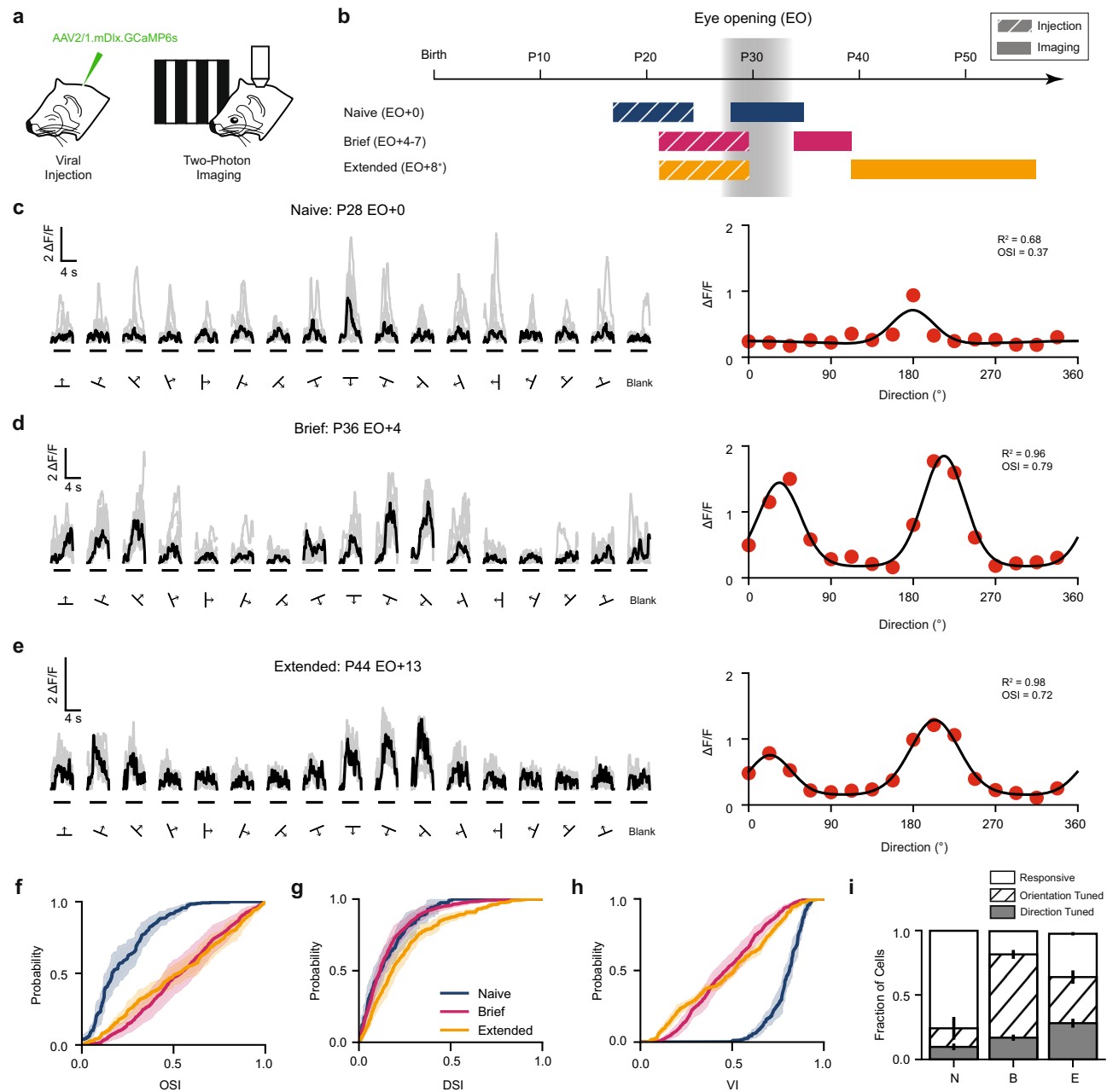

**Fig. 1 Orientation selectivity strengthens in interneurons after eye opening. a** Two-photon imaging of viral expression of GCaMP6s under the mDlx enhancer in ferret visual cortex. The ferret images were created by the authors and published in Smith, G.B., Sederberg, A., Elyada, Y.M., Van Hooser, S.D., Kaschube, M., and Fitzpatrick, D., The development of cortical circuits for motion discrimination. *Nat. Neurosci.* 18, 252–261 (2015). **b** Three age groups, Naive (EO + 0), Brief experience (EO + 4–7), and Extended experience (EO + 8+). **c** Example cellular responses to square drifting gratings (left, black traces denote median response and gray traces denote single trials) and responsive tuning curve (right) for a Naive animal. **d, e** same as (**c**) but for a Brief and Extended experience animal respectively. **f–h** Cumulative distributions of orientation selectivity index (**f**), direction selectivity (**g**), and variability index (**h**) for Naive (blue, $n = 7$), Brief (magenta, $n = 6$), and Extended (gold, $n = 5$) experience animals. Shaded regions denote SEM. **i** Fraction of cells that are responsive (white), orientation selective (hatched), and direction selective (gray) for Naive (N, $n = 7$), Brief (B, $n = 6$), and Extended (E, $n = 5$) experience animals. Error bars denote SEM. Source data are provided as a Source Data file.

reflects activity indistinguishable from spontaneous activity. For all age groups, GABA-INs had SAIs significantly larger than zero, (Supplementary Fig. 1a, SAI Naive $1.43 \pm 0.05$, $p < 0.0001$, $n = 233$ cells; Brief $1.00 \pm 0.03$, $p < 0.0001$, $n = 966$ cells; Extended $0.17 \pm 0.04$, $p < 0.0001$, $n = 326$ cells; Mean ± SEM, Students' $t$ test). Thus, even at the non-preferred orientation, visual stimulation drives GABA-IN activity.

The degree to which visual stimulation drives activity to the non-preferred orientation decreased with experience (SAI Naive

$1.43 \pm 0.05$ vs. Brief $1.00 \pm 0.03$, $p < 0.0001$, $n = 233$ vs. 966 cells; Naive $1.43 \pm 0.05$ vs. Extended $0.17 \pm 0.04$, $p < 0.0001$, $n = 233$ vs. 326 cells; Mean ± SEM, bootstrap test). This raises the possibility that visually-evoked activity to the non-preferred orientation contributes to the weak selectivity of GABA-INs early in development. If this is the case, we would expect to see an inverse relationship between OSI and SAI—as the response to the non-preferred orientation weakens (lower SAI), orientation selectivity increases (higher OSI). Indeed, GABA-INs displayed a weak but

significant negative correlation between OSI and SAI in Naive and Brief, but not Extended experience animals (Supplementary Fig. 1b, Naive $r = -0.28$, $p < 0.0001$, $n = 233$ cells; Brief $r = -0.25$, $p < 0.0001$, $n = 966$ cells; Extended $r = -0.05$, $p = 0.3730$, $n = 326$ cells; Pearson's r). Therefore, visual responsiveness to the non-preferred orientation contributes to weak orientation selectivity of GABA-INs in early development.

A property of excitatory neurons that undergoes an experience-dependent change following eye opening is the binocular alignment of orientation preferences. Before eye opening, excitatory neurons often have mismatched preferences for each eye[4]. We thought it would be important to determine whether monocular orientation preference mismatch contributed to the weak orientation selectivity in the Naive group since these initial experiments were conducted using binocular stimulation. Using monocular stimulation, we found that almost all GABA-INs in the Naive group responded to monocular stimulation of the contralateral and ipsilateral eye (Supplementary Fig. 2d, h; Fraction of responsive cells, Naive contralateral $0.994 \pm 0.003$, ipsilateral $0.992 \pm 0.03$, $n = 7$, Mean ± SEM). Like binocular responses, monocular responses were not well-tuned to orientation or direction (Supplementary Fig. 2a, b, e, f. Fraction of orientation tuned cells Naive contralateral $0.144 \pm 0.042$, ipsilateral $0.192 \pm 0.085$; Fraction of direction tuned cells Naive contralateral $0.072 \pm 0.03$, ipsilateral $0.044 \pm 0.016$; $n = 7$, Mean ± SEM), supporting our conclusion that GABA-INs in visually naive animals lack orientation selectivity for monocular and binocular stimulation.

The high percentage of binocular GABA-INs in the Naive group led us to wonder whether this property was maintained and whether GABA-INs exhibited an initial mismatch in orientation selectivity like that found for excitatory neurons. We found that GABA-INs maintain a high degree of binocularity across all groups examined, with no significant shift in monocularity (Supplementary Fig. 2i; |ODI| Naive $0.14 \pm 0.02$ vs. Brief $0.16 \pm 0.02$, $p = 0.3216$, $n = 7$ vs. 6; Naive $0.14 \pm 0.02$ vs. Extended $0.14 \pm 0.01$, $p = 0.8352$, $n = 7$ vs. 5; Mean ± SEM, bootstrap test). Moreover, there was no significant difference in the monocular orientation preference mismatch for experienced animals (Supplementary Fig. 2l; Ipsilateral vs. Contralateral, Brief $8.54 \pm 1.17$ vs. Extended $6.26 \pm 1.48$, $p = 0.2116$; $n = 6$ vs. 5, bootstrap significance), or in the mismatch of monocular and binocular orientation preferences (Supplementary Fig. 2j, k; Binocular vs. Contralateral, Brief $4.98 \pm 0.77$ vs. Extended $5.15 \pm 0.60$, $p = 0.8418$; Binocular vs. Ipsilateral, Brief $7.63 \pm 1.01$ vs. Extended $5.13 \pm 0.44$, $p = 0.0560$; $n = 6$ vs. 5; Mean ± SEM, bootstrap significance).

Together, these results indicate that inhibitory and excitatory neurons have different developmental progressions. Excitatory neurons exhibit robust orientation selectivity before eye opening, and this is accompanied by ocular differences in orientation preferences. Subsequent experience drives binocular alignment of orientation preference and enhances the selectivity of excitatory neurons[1,4]. In contrast, GABA-INs are weakly orientation-selective before eye opening and increases in orientation selectivity arise concurrently with binocularly aligned preferences (without a misalignment phase) after eye opening. The nature of the synaptic changes that underlie the emergence of tuned responses in inhibitory neurons remains to be determined, but GABA-INs likely derive orientation-selective responses when the balance of their excitatory inputs arises from a network that has already achieved binocularly aligned responses.

**Developmental emergence of orientation-selective signals in GABA-IN populations**. Although we found that GABA-INs in Naive animals are not well-tuned for orientation, we wondered

whether there was enough reliability in responses to support a population-level representation. To assess this, we measured the trial-to-trial pattern correlation (Fig. 2a) of stimulus-evoked responses for the entire population of GABA-INs across all orientations and then compared matched and orthogonal stimuli. In Naive and experienced animals, trial-to-trial correlations for matched stimuli were significantly higher than orthogonal stimuli (Fig. 2c, Naive, matched $0.59 \pm 0.02$ vs. orthogonal $0.50 \pm 0.04$, $p = 0.0316$, $n = 7$; Brief matched $0.71 \pm 0.02$ vs. orthogonal $0.02 \pm 0.07$, $p = 0.0006$, $n = 6$; Extended matched $0.76 \pm 0.02$ vs. orthogonal $0.09 \pm 0.02$, $p < 0.0001$, $n = 5$; Mean ± SEM, Student's paired $t$ test). Although the difference in trial-to-trial correlations was significant for Naive animals, the difference was a fraction of that found in experienced animals, such that pattern correlations were significantly higher for matched stimuli (Naive $0.59 \pm 0.02$ vs. Brief $0.71 \pm 0.02$, $p = 0.0004$, $n = 7$ vs. 6; Naive $0.59 \pm 0.02$ vs. Extended $0.76 \pm 0.02$, $p = 0.0012$, $n = 7$ vs. 5; Mean ± SEM, bootstrap test), and significantly lower for orthogonal stimuli (Naive $0.50 \pm 0.04$ vs. Brief $0.03 \pm 0.07$, $p = 0.0016$, $n = 7$ vs. 6; Naive $0.50 \pm 0.04$ vs. Extended $0.09 \pm 0.02$, $p = 0.0010$, $n = 7$ vs. 5; Mean ± SEM, bootstrap test). These data demonstrate that oriented stimuli evoke responses in GABA-INs at eye opening, but the response pattern is only weakly related to stimulus orientation and is similar across stimuli.

To understand the functional implications of these changes in stimulus-evoked responses, we compared the accuracy of GABA-INs from naive and experienced animals in discriminating orientations by using a template matching decoder[4,18]. Briefly, we generated a pattern template set for each orientation from evoked activity patterns and compared them to individual trials to predict the stimulus (Fig. 2d). Across all cohorts, our decoder predicted stimuli better than chance (Fig. 2e, Decoding Accuracy for 40 Cells compared to shuffle, Naive $0.136 \pm 0.019$, $p = 0.0094$, $n = 7$; Brief $0.578 \pm 0.053$, $p = 0.0002$, $n = 6$; Extended $0.720 \pm 0.051$, $p = 0.0002$, $n = 5$; chance level across all groups was 0.06, Mean ± SEM, Student's $t$ test). Naive animals, however, decoded stimuli at a lower rate compared to experienced animals (decoding accuracy for 40 cells, Naive $0.136 \pm 0.019$ vs. Brief $0.578 \pm 0.053$, $p = 0.0006$, $n = 7$ vs. 6; Naive $0.136 \pm 0.019$ vs. Extended $0.720 \pm 0.051$, $p < 0.0001$, $n = 7$ vs. 5; Mean ± SEM, bootstrap test). Additionally, when we assessed the change in decoding accuracy for templates including one cell compared to two, experienced animals showed a greater improvement in accuracy (Fig. 2f, Naive $0.010 \pm 0.003$ vs. Brief $0.062 \pm 0.008$, $p = 0.0002$, $n = 7$ vs. 6; Naive $0.010 \pm 0.003$ vs. Extended $0.057 \pm 0.005$, $p = 0.0004$, $n = 7$ vs. 5; Mean ± SEM, bootstrap test), suggesting cells carry more information later in development. In sum, these data demonstrate dramatic changes in the population response properties of GABA-INs after the onset of visual experience that endows them with a greater capacity to discriminate stimulus orientation.

**Clustered responses and modular patterns emerge after visual experience**. We wondered how changes in the responses of GABA-INs related to the modular organization of orientation preferences. Modular organization has been observed in both mature and naive excitatory as well as mature inhibitory networks in the ferret[4,6,8]. To assess the state of the modular organization, we used widefield epifluorescence calcium imaging to visualize the response of GABA-INs at the millimeter scale (Fig. 3a). In experienced animals, we observed clustered responses to oriented stimuli which gave rise to smoothly varying orientation preference maps (Fig. 3b, Supplementary Fig. 2a). Orientation preference maps, however, were not present in Naive animals and a smaller proportion of visually responsive pixels had significant

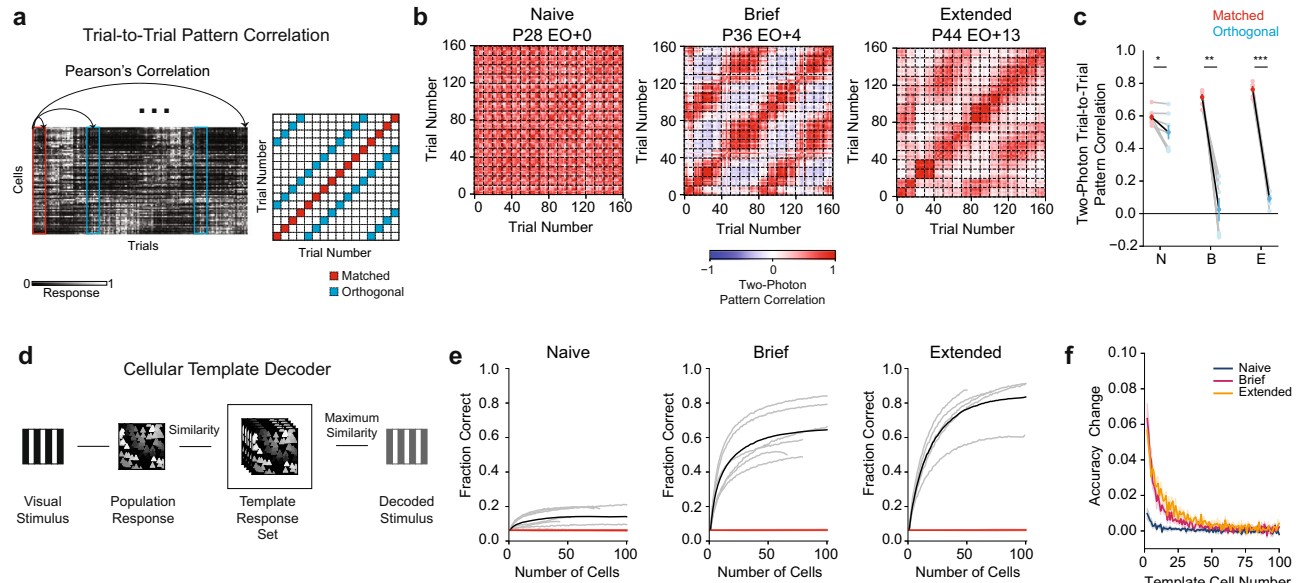

**Fig. 2 Population responses of GABA-INs to orientation become more differentiable after visual experience. a** Trial-to-trial pattern correlation matrices were generated by comparing population trial responses across stimulus conditions. Trials were sorted by direction, such that each group of 10 trials corresponds to the repeated presentation of a stimulus direction. Sixteen directions were presented equally spaced between 0 and 337.5 degrees. **b** Example trial-to-trial pattern correlation matrices for a Naive (left), Brief (middle), and Extended (right) experience animal. Dashed lines denote groups of the same stimulus direction. **c** Comparisons of trial-to-trial pattern correlation for matched (red) and orthogonal (blue) stimuli across age groups (Naive: N ($p = 0.0316$, $n = 7$), Brief: B ($p = 0.0006$, $n = 6$), Extended: E ($p < 0.0001$, $n = 5$)). Two-tailed bootstrap test. **d** Schematic of the template decoding algorithm. **e** Decoding fraction versus number of cells used in the template decoder for Naive (left, $n = 7$), Brief (middle, $n = 6$), and Extended (right, $n = 5$) animals. Results for individual animals (gray), grouped average (black), and shuffle (red). **f** Change in decoding accuracy versus the cell number in the decoder for Naive (blue, $n = 7$), Brief (magenta, $n = 6$), and Extended experience (gold, $n = 5$). Shaded regions denote SEM. *$p < 0.05$; **$p < 0.005$; ***$p < 0.0005$, two-tailed bootstrap test. Source data are provided as a Source Data file.

orientation tuning (Fig. 3b, d; Naive $0.32 \pm 0.12$ vs. Brief $0.88 \pm 0.02$, $p = 0.0076$, $n = 5$ vs. 4; Naive $0.32 \pm 0.12$ vs. Extended $0.94 \pm 0.03$, $p = 0.0050$, $n = 5$ vs. 5; Mean $\pm$ SEM, bootstrap test).

The absence of orientation preference maps in Naive animals does not rule out the presence of modular patterns of correlated activity in inhibitory networks independent of selectivity for orientation. Therefore, we computed correlation maps using pairwise pixel response correlations for drifting gratings, allowing us to assess the structure of evoked responses independent of the visual stimuli presented. Consistent with our previous results, modular patterns were apparent in the response correlation maps for experienced animals (Fig. 3c). Average correlation and distance displayed a complex relationship, deviating from linearity (Fig. 3e, Brief linear regression $R^2 = 0.33 \pm 0.03$, Extended linear regression $R^2 = 0.35 \pm 0.05$). In contrast, we did not observe modular clustering in Naive animals (Fig. 3c), and correlation strength decreased almost linearly with distance (Fig. 3e, Naive linear regression $R^2 = 0.88 \pm 0.06$).

To further quantify the modularity of the correlation maps, we modified a method for measuring the column spacing (wavelength) of modular networks[19]. We observed comparable wavelengths for excitatory neurons and GABA-INs in experienced animals (Supplementary Fig. 3b; Brief excitatory $0.736 \pm 0.032$ mm vs. GABA-INs $0.770 \pm 0.053$ mm, $n = 6$ vs. 4, $p = 0.5120$; Extended excitatory $0.797 \pm 0.057$ mm vs. GABA-INs $0.844 \pm 0.041$ mm, $n = 5$ vs. 5, $p = 0.4730$; Mean $\pm$ SEM, bootstrap test). In contrast, GABA-INs in Naive animals had a significantly larger wavelength (Naive excitatory $0.779 \pm 0.020$ vs. GABA-INs $0.924 \pm 0.026$, $n = 7$ vs. 5, $p = 0.0048$, Mean $\pm$ SEM, bootstrap test), suggesting that responses are not as well associated with excitatory modular networks. Importantly, modular organization was still apparent and wavelengths were comparable when regions of interest for Naive animals were used

to mask Brief experience correlation patterns (Supplementary Fig. 3c).

To control for uniform patterns of correlation, we computed wavelengths for uniform correlation patterns within our regions of interest. Measured wavelengths for animals with brief experience were significantly smaller than uniform correlation wavelengths (Brief $0.768 \pm 0.033$ mm vs. Uniform $0.944 \pm 0.035$ mm, $n = 4$, $p = 0.0105$, Mean $\pm$ SEM, Student's paired $t$ test), but they were not significantly different for Naive animals (Naive $0.890 \pm 0.023$ mm vs. Uniform $0.893 \pm 0.035$ mm, $n = 5$, $p = 0.9270$, Mean $\pm$ SEM, Student's paired $t$ test). Therefore, in Naive animals, measured wavelengths may reflect uniform correlation across the field-of-view and a lack of modular organization. Thus, the difference in local clustering of correlations and wavelengths observed before eye opening suggests naive inhibitory networks may lack modular organization at the millimeter-scale for evoked activity, and after a week of visual experience modular organization of responses and orientation preferences emerge.

While widefield imaging has shown that inhibitory networks lack large-scale organization for evoked activity in Naive animals, cellular-scale organization could still be present. For example, the lack of modular structure in Naive animals could arise from the uniform activation or variable 'salt-and-pepper' activation of cells. To assess the cellular response patterns, we used two-photon imaging to measure the cellular pairwise correlations of evoked responses, independent of stimulus identity for short-range ($<100 \mu m$) and long-range ($300–400 \mu m$) cell pairs. Across all groups, cellular pairwise response correlations for short-range pairs were significantly higher than for long-range (Fig. 3g, Naive short $0.49 \pm 0.07$ vs. long $0.36 \pm 0.08$, $p = 0.0003$, $n = 7$; Brief short $0.43 \pm 0.03$ vs. long $0.04 \pm 0.03$, $p = 0.0009$, $n = 6$; Extended short $0.30 \pm 0.04$ vs. long $0.11 \pm 0.03$, $p = 0.0004$, $n = 5$; Mean $\pm$ SEM, Student's paired $t$ test), suggesting that "salt-and-

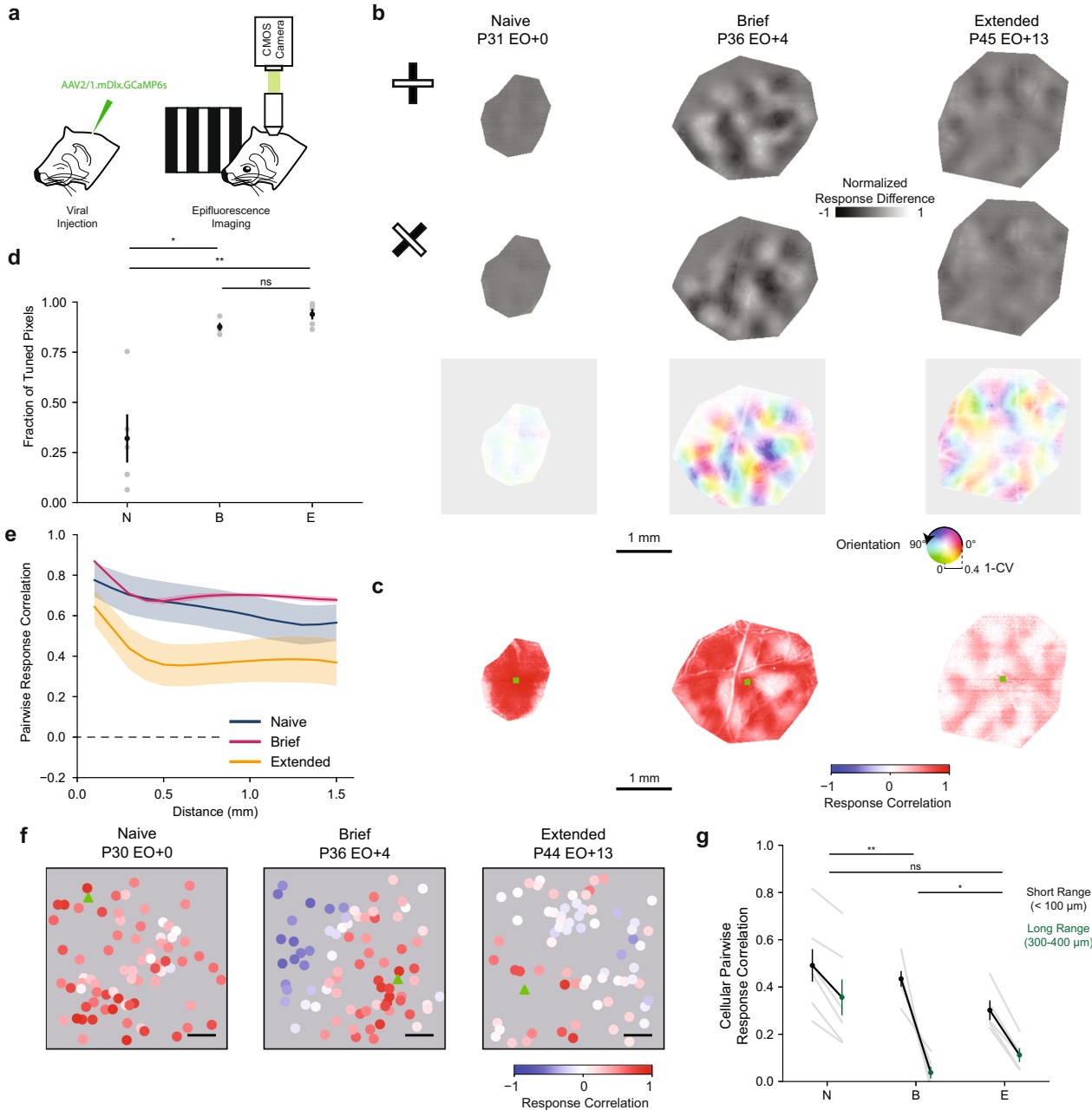

**Fig. 3 Clustered responses emerge across development in inhibitory networks. a** Widefield epifluorescence calcium imaging of GCaMP6s under the mDlx enhancer in ferret visual cortex The ferret images were created by the authors and published in Smith, G.B., Sederberg, A., Elyada, Y.M., Van Hooser, S.D., Kaschube, M., and Fitzpatrick, D., The development of cortical circuits for motion discrimination. *Nat. Neurosci.* 18, 252–261 (2015). **b** Difference Maps for orthogonal stimuli (top, middle) and orientation preference maps (bottom) for a Naive (left), Brief (Middle), and Extended (right) experience animal. **c** Example pairwise pixel response correlation maps for a Naive (left), Brief (Middle), and Extended (right) experience animal. Green squares identify the seed positions. **d** Fraction of pixels that are significantly orientation tuned compared to shuffle (Naive $n = 5$, Brief $n = 4$, Extended $n = 5$; Naive vs. Brief $p = 0.0076$, Naive vs. Extended $p = 0.0050$, Brief vs. Extended $p = 0.0862$). Mean ± SEM, Two-tailed bootstrap test. **e** Mean distance dependence of pairwise pixel response correlations for Naive (blue, $n = 5$), Brief (magenta, $n = 4$), and Extended (gold, $n = 5$) experience animals. Shaded regions denote the SEM. **f** Example cellular pairwise correlation patterns for a Naive (left), Brief (middle), and Extended experience (right) animals. The seed cell is denoted by the green triangle and scale bar denotes 100 μm. **g** Comparisons of the average short-range (black, <100 μm) and long-range (green, 300–400 μm) cellular pairwise correlations for Naive (N, $p = 0.0003$, $n = 7$), Brief (B, $p = 0.0050$, $n = 6$), and Extended (E, $p = 0.0004$, $n = 5$) experience animals. Mean ± SEM. *$p < 0.05$, **$p < 0.005$, ns not significant, Two-tailed Student's paired $t$ test. Source data are provided as a Source Data file.

pepper" response patterns are not typical. When comparing the relative difference between short- and long-range correlations, however, Naive animals had a significantly lower difference in correlation compared to Brief experience animals (Fig. 3g, Naive 0.14 ± 0.02 vs. Brief 0.40 ± 0.06, $p = 0.0024$, $n = 7$ vs. 6, Mean ±

SEM, bootstrap test). These findings were consistent with patterns of activity, as responses for Naive animals often showed widespread activity, which was not observed in experienced animals (Supplementary Fig. 4b, c). Importantly, cortical expansion cannot account for increased clustering between Naive

and Brief experience animals, as it would predict a reduction in clustering.

We also observed a significant reduction in the clustering between animals with Brief and Extended experience (Fig. 3g, Brief $0.40 \pm 0.06$ vs. Extended $0.19 \pm 0.02$, $p = 0.0106$, bootstrap test). Loss of responsivity resulting from the overexpression of GCaMP6s is likely not the cause, as a large fraction of GABA-INs remained visually responsive (Fig. 1i). Lower response correlations could result from cortical expansion and/or from diversification of neuronal preferences along other relevant dimensions (such as spatial or temporal frequency). Diversification of neuronal preferences, which is theorized to contribute to the efficient coding of stimuli[20], results in increased cellular responses heterogeneity and could reflect optimization of visual representations in the second week of visual experience. Our findings suggest that correlated responses in naive inhibitory networks are only modestly greater for neighboring cells than those at greater distances and widespread, correlated activity dominates responses. This correlated activity may mask modular network structure. After eye opening, the correlation of neighboring cells remains high and long-range correlations decrease, resulting in observable patchy, clustered patterns of activity (Supplementary Fig. 4b).

**Development of orientation selectivity in GABA-INs depends on visual experience.** Visual experience is a necessary contributor to the normal maturation of excitatory networks and delayed experience results in deficits to orientation selectivity[1,4]. Therefore, we wondered whether GABA-IN development requires visual experience. We performed binocular deprivation in a cohort of animals (BD) by eyelid suturing for 5–8 days, depriving them of structured visual experience (Fig. 4a). We then assessed the responses of cells to oriented stimuli after manually opening the eyes. GABA-INs in BD animals had robust visually-evoked activity, and the amplitude of these responses was not strongly dependent on the orientation or direction of the visual stimulus (Fig. 4b, c). When compared to Brief experience animals, BD animals displayed significantly lower orientation selectivity (OSI BD $0.22 \pm 0.04$ vs. Brief $0.54 \pm 0.06$, $p = 0.009$, $n = 4$ vs. 6; Mean $\pm$ SEM, bootstrap test) and comparable direction selectivity (DSI BD $0.13 \pm 0.03$ vs. Brief $0.19 \pm 0.03$, $p = 0.5196$, $n = 4$ vs. 6; Mean $\pm$ SEM, bootstrap test). BD animals also showed higher response variability (VI BD $0.75 \pm 0.01$ vs. Brief $0.48 \pm 0.04$, $p = 0.0038$, $n = 4$ vs. 6; Mean $\pm$ SEM, bootstrap test). Together, these data suggest that the functional properties of GABA-INs require visual experience to reach mature levels.

To test the necessity of visual experience over the first week following normal eye opening, we performed binocular eyelid suturing for 9 days, manually opened the eyes, and assessed visual responses after 8–15 days of visual experience (Fig. 4a, b, Recovery). Interneurons in Recovery animals responded robustly to oriented stimuli (Fig. 4d) and had stronger orientation selectivity than BD animals (Fig. 4e, OSI BD $0.22 \pm 0.04$ vs. Recovery $0.40 \pm 0.03$, $p = 0.0170$, $n = 4$ vs. 4, bootstrap test). Direction selectivity, however, did not significantly change (Fig. 4f, DSI BD $0.13 \pm 0.03$ vs. Recovery $0.18 \pm 0.03$, $p = 0.2350$, $n = 4$ vs. 4, bootstrap test). In addition, response variability decreased with experience (Fig. 4g, VI BD $0.75 \pm 0.01$ vs. Recovery $0.50 \pm 0.02$, $p = 0.0042$, $n = 4$ vs. 4, bootstrap test). When comparing Recovery and Extended experience animals, however, we found no significant difference in orientation selectivity, direction selectivity, or cellular variability (OSI, Recovery $0.41 \pm 0.03$ vs. Extended $0.52 \pm 0.05$, $p = 0.0650$; DSI, Recovery $0.18 \pm 0.03$ vs. Extended $0.24 \pm 0.03$, $p = 0.1216$; VI, Recovery $0.50 \pm 0.02$ vs. Extended $0.48 \pm 0.02$, $p = 0.3648$; $n = 4$ vs. 5, bootstrap test).

In sum, these data suggest that the maturation of orientation selectivity and reduction in cellular variability of GABA-INs depend on visual experience but can occur beyond the first week following normal eye opening.

**Development of orientation-specific population responses requires visual experience.** Next, we assessed the correspondence of evoked population patterns to stimuli, by assessing the trial-to-trial pattern correlation for matched and orthogonal orientations (Fig. 5a). BD animals displayed a high similarity for matched and orthogonal orientations (Fig. 5b, BD Matched $0.59 \pm 0.02$ vs. Orthogonal $0.49 \pm 0.05$, $p = 0.0718$, $n = 4$, Mean $\pm$ SEM, Student's paired $t$ test). The response correlations for matched and orthogonal orientations were not significantly different from Naive animals (BD matched $0.59 \pm 0.03$ vs. Naive $0.59 \pm 0.03$, $p = 0.9562$; BD orthogonal $0.49 \pm 0.05$ vs. Naive orthogonal $0.50 \pm 0.04$, $p = 0.8114$, $n = 4$ vs. 7, Mean $\pm$ SEM, Student's paired $t$ test). Thus, inhibitory networks in BD animals lack a population-level representation of orientation, and visual experience is necessary for the formation of orientation-specific responses.

Orientation selectivity of GABA-INs requires visual experience that can be delayed beyond the period of normal onset, but we wondered if the population responses would be normal in recovery animals. In Recovery animals, response pattern correlations were higher for matched stimuli (Matched Stimuli Pattern Correlation, Recovery $0.81 \pm 0.02$ vs. BD $0.59 \pm 0.03$, $p = 0.0064$, $n = 4$ vs. 4; Mean $\pm$ SEM, bootstrap test) and lower for orthogonal stimuli (Orthogonal Stimuli Pattern Correlation, Recovery $0.25 \pm 0.02$ vs. BD $0.49 \pm 0.05$, $p = 0.0108$, $n = 4$ vs. 4; Mean $\pm$ SEM, bootstrap test). We observed similar differences when comparing Recovery and Naive animals (Recovery matched $0.81 \pm 0.02$ vs. Naive $0.59 \pm 0.02$, $p = 0.0014$; Recovery orthogonal $0.2503 \pm 0.02$ vs. Naive $0.50 \pm 0.04$, $p = 0.0048$; $n = 7$ vs. 4, Mean $\pm$ SEM, bootstrap test). Age-matched animals with normal visual experience also displayed comparable, high similarity of response patterns for matched stimuli, and lower similarity for orthogonal stimuli (matched Recovery $0.81 \pm 0.02$ vs. Extended $0.76 \pm 0.02$, $p = 0.1118$; orthogonal Recovery $0.25 \pm 0.02$ vs. Extended $0.09 \pm 0.02$, $p = 0.0076$; $n = 5$ vs. 4, Mean $\pm$ SEM, bootstrap test). These data suggest that orientation-related inhibitory network pattern development is resilient to the delayed onset of experience.

We further assessed the relationship of stimulus-evoked responses to orientation by testing a template decoder for BD and Recovery animals (Fig. 5c). Consistent with our trial-to-trial pattern correlation findings, BD animals significantly underperformed as compared to Recovery animals (Decoding Accuracy for 40 cell template BD $0.185 \pm 0.023$ vs. Recovery $0.713 \pm 0.031$, $p = 0.8950$, $n = 4$ vs. 4, Mean $\pm$ SEM, bootstrap test). Recovery animals also showed marked improvement in decoding of orientation even for templates using small numbers of cells (Change in decoding accuracy for 2nd cell, BD $0.014 \pm 0.03$ vs. Recovery $0.053 \pm 0.005$, $p = 0.0038$, $n = 4$ vs. 4, Mean $\pm$ SEM, bootstrap test). Thus, visual experience is necessary for the development of orientation-related responses of GABA-INs, improving the discriminability of orthogonal stimuli responses and response reliability.

While Recovery animals showed improvement in orientation-related response discriminability, it remains possible that the inhibitory networks of these animals may have an impaired encoding of orientations. Recovery animals, however, did not significantly underperform when compared to age-matched animals with normal experience (Decoding Accuracy for 40 cell template Recovery $0.713 \pm 0.031$ vs. Extended $0.720 \pm 0.051$,

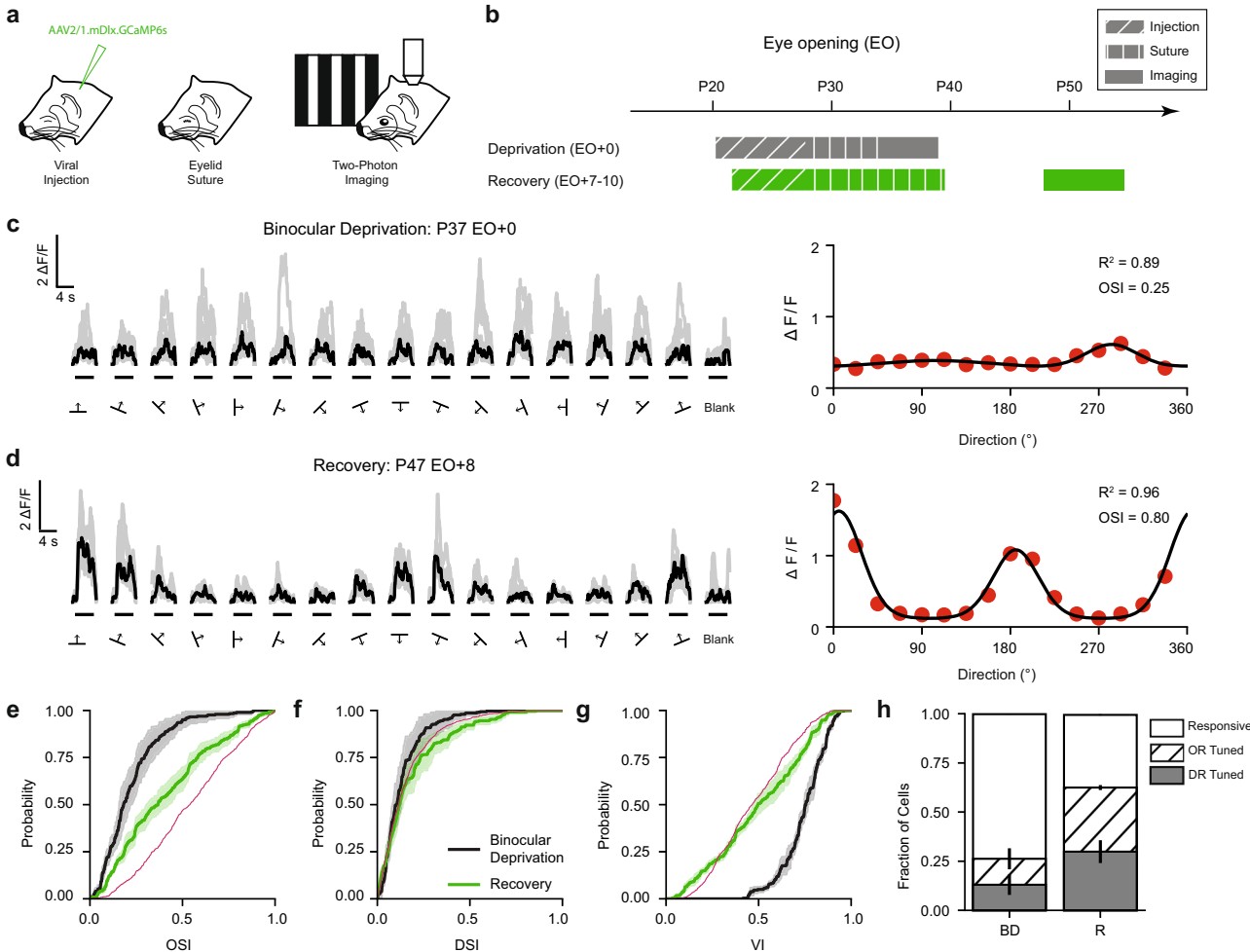

**Fig. 4 Development of orientation preference in interneurons requires visual experience. a** Two-photon imaging of viral expression of GCaMP6s under the mDlx enhancer in ferret visual cortex after binocular eyelid suture. The ferret images were created by the authors and published in Smith, G.B., Sederberg, A., Elyada, Y.M., Van Hooser, S.D., Kaschube, M., and Fitzpatrick, D., The development of cortical circuits for motion discrimination. *Nat. Neurosci.* 18, 252–261 (2015). **b** Two cohorts, Binocular Deprivation (EO + 0, 5–8 days of deprivation) and Recovery (EO + 8–15, 9 days of deprivation) were evaluated for GABA-IN responses to oriented stimuli. **c** Example cellular response to drifting gratings (left, black traces denote median response and gray traces denote single trials) and response tuning curve (right) for a binocularly deprived animal. **d** Example cellular response to drifting gratings (left) and response tuning curve (right) for a recovery animal. **e–g** Cumulative distributions of orientation selectivity index (**e**), Cumulative distributions of direction selectivity (**f**), and cumulative distributions of variability index (**g**) for BD (black, $n = 4$) and Recovery (green, $n = 4$) animals. Shaded regions denote SEM. Magenta line is provided as reference for Brief experience (see Fig. 1). **h** Fraction of cells that are responsive (white), orientation selective (hatched), and direction selective (gray) for Binocular Deprivation (BD, $n = 4$) and Recovery (R, $n = 4$) animals. Error bars denote SEM. Source data are provided as a Source Data file.

$p = 0.8950$, $n = 5$ vs. 4; Mean ± SEM, bootstrap), and the change in decoding accuracy for small numbers of cells was also comparable (Fig. 5d, Change in decoding accuracy for 2nd cell, Recovery $0.053 ± 0.005$ vs. Extended $0.057 ± 0.005$, $p = 0.5606$, $n = 5$ vs. 4, Mean ± SEM, bootstrap test) suggesting that there is not a strict critical period for the experience-dependent development of orientation-specific population responses for inhibitory networks.

**Developmental reduction in non-modular responses does not require visual experience**. These results indicate that experience plays a role in the development of orientation-selective responses in GABA-INs, and one might expect that experience plays a similar role in the development of modular spatial organization for GABA-INs. First, we assessed whether large-scale organization

was observable in BD and Recovery animals, using widefield epifluorescence imaging. Patterns of activity in both BD and Recovery animals displayed clustered, modular organization (Supplementary Fig. 5a, b). We further assessed the modularity of the pairwise pixel response correlations by computing the column spacing. Both BD and Recovery animals had comparable wavelengths to excitatory neurons (Supplementary Fig. 5c; BD excitatory $0.867 ± 0.065$ mm vs. GABA-INs $0.897 ± 0.064$ mm, $p = 0.6974$, $n = 4$ vs. 4; Recovery excitatory $0.796 ± 0.046$ vs. GABA-INs $0.798 ± 0.031$ mm, $p = 0.9618$, $n = 5$ vs. 4; Mean ± SEM, bootstrap), and wavelengths for BD animals were comparable to GABA-INs in animals with normal visual experience (BD $0.897 ± 0.064$ mm vs. Brief $0.770 ± 0.053$ mm, $p = 0.1390$, $n = 4$ vs. 4; Mean ± SEM, bootstrap). Thus, modular organization in GABA-INs appears independent of visual experience over the first week following normal eye opening.

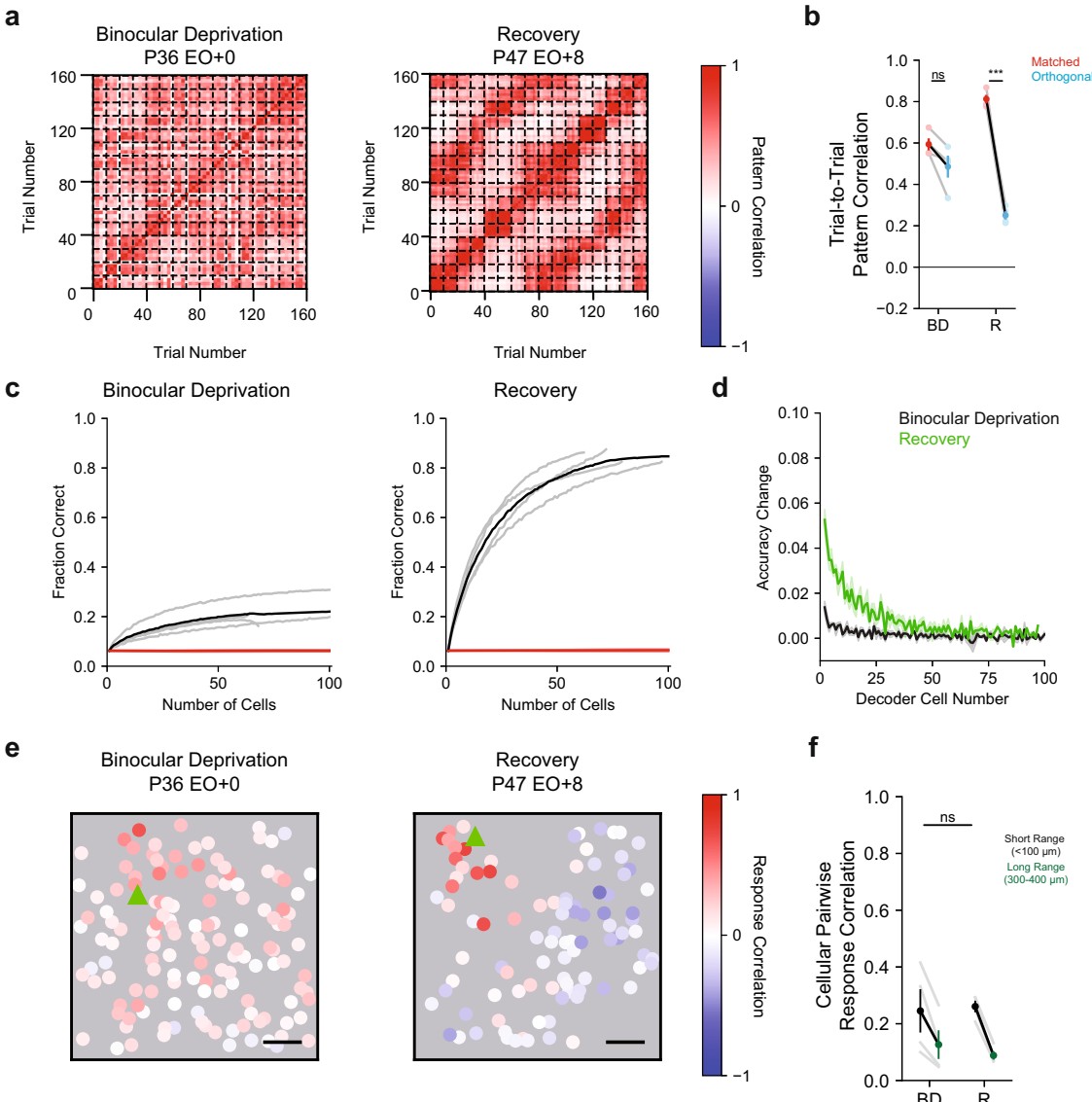

**Fig. 5 Reduction in spatially uniform responses occurs without visual experience. a** Trial-to-trial pattern correlations for a Binocular Deprivation (left) and Recovery (right) animal. Trials were sorted by direction, such that each group of 10 trials corresponds to the repeated presentation of a stimulus direction. Sixteen directions were presented equally spaced between 0 and 337.5 degrees. Dashed lines denote groups. **b** Comparison of matched (red) and orthogonal (blue) trial-to-trial pattern correlations for Binocular Deprivation ($p = 0.0718$, $n = 4$) and Recovery ($p = 0.0003$, $n = 4$) animals. Individual animals are shown as gray and error bars denote SEM. Two-tailed paired Student's $t$ test. **c** Template decoding accuracy for Binocular Deprivation ($n = 4$) and Recovery ($n = 4$) animals for different template sizes. Red line indicates chance decoding levels, black shows average decoding level, and gray lines denote individual animals. **d** Change in decoder accuracy for the nth cell included in the template decoder for Binocular Deprivation (black, $n = 4$) and Recovery (green, $n = 4$) animals. Shaded regions denote SEM. **e** Example cellular pairwise correlations for Binocular Deprivation (left) and Recovery (right) animals. The seed cell is denoted by the green triangle and scale bar denotes 100 μm. **f** Binocular Deprivation ($n = 4$) and Recovery ($n = 4$) differences of cellular pairwise response correlations for short- (<100 μm) and long-range (300–400 μm) pair are similar. ***$p < 0.0005$, ns not significant. Source data are provided as a Source Data file.

How well is the large-scale organization of GABA-INs reflected in the cellular response? We assessed the spatial organization of cellular response correlations (Fig. 5e, Supplementary Fig. 5d–f) and found that both BD and Recovery animals demonstrated higher correlations for short-range (<100 μm) compared to long-range (300–400 μm) cell pairs (Fig. 5f, BD short $0.25 \pm 0.08$ vs. long $0.13 \pm 0.05$, $p = 0.0357$, $n = 4$; Recovery short $0.26 \pm 0.02$ vs. long $0.09 \pm 0.02$, $p = 0.0028$, $n = 4$; Mean ± SEM, Student's paired $t$ test). The difference in correlations for short- and long-range cell pairs were like animals with normal extended experience (BD $0.12 \pm 0.03$ vs. Extended $0.19 \pm 0.02$, $p = 0.0630$, $n = 4$ vs. 4, Mean ± SEM, bootstrap test). Importantly, correlations for both short- and long-range cell pairs were systematically lower for BD animals when compared to Naive animals (Naive short $0.49 \pm 0.07$ vs. BD short $0.25 \pm 0.08$, $p = 0.0392$; Naive long $0.36 \pm 0.08$ vs. BD long $0.13 \pm 0.05$, $p = 0.0448$; $n = 4$ vs. 7, Mean ± SEM, bootstrap test). However, BD animals had a significantly lower correlation for neighboring GABA-INs compared to age-matched animals with Brief experience (BD short $0.25 \pm 0.08$ vs. Brief short $0.43 \pm 0.03$, $p = 0.0392$; BD long $0.13 \pm 0.05$ vs. Brief long $0.04 \pm 0.03$, $p = 0.1012$; $n = 4$ vs. 6, Mean ± SEM, bootstrap test). Thus, the widespread, correlated activity that dominates responses in Naive animals was reduced in BD animals, but short-range correlations were impaired. In

Recovery animals, the subsequent experience did not affect the difference in short- and long-range correlations (BD $0.12 \pm 0.03$ vs. Recovery $0.17 \pm 0.02$, $p = 0.1572$; $n = 4$ vs. 4, Mean ± SEM, bootstrap test), suggesting delayed visual experience does not alter the degree to which responses to oriented stimuli cluster. Together, our findings demonstrate that the reduction of widespread, correlated activity and subsequent unmasking of clustered activity in inhibitory networks can develop in an experience-independent manner over the developmental period spanning the first week of normal visual experience. However, the maintenance of short-range correlations early in development and orientation-associated modular responses requires visual experience.

## Discussion

In summary, we have demonstrated that before visual experience, interneurons in the ferret visual cortex show poor orientation tuning because of variable responses. These variable responses manifest as a global pattern of correlated activity that recruits all interneurons independent of the visual stimulus orientation. As little as 1 week of visual experience, however, suffices to drive the development of strong orientation tuned responses, reduction in widespread, correlated activity, and the association of modular patterns to stimuli. Furthermore, interneurons are resilient to delayed experience onset- maintaining the ability to develop orientation tuning beyond the first week of normal visual experience. Finally, the emergence of modular inhibitory responses does not require visual experience, as we observed single-trial modular response patterns in animals after a week of binocular deprivation. However, the reliable association of these patterns to oriented stimuli requires visual experience, as binocularly deprived animals displayed deficits in orientation selectivity and network pattern differentiability.

Excitatory and inhibitory neurons in the visual cortex around eye opening undergo distinct developmental trajectories (Fig. 6). At eye opening, excitatory networks display modular patterns of activity for both spontaneous and evoked activity, and initial cellular orientation preferences are already established[4,6,7]. A recent study suggested that GABA-INs also display modular patterns of spontaneous activity before eye opening[21]. In contrast, we show that modular patterns are not evident for visually-evoked activity and orientation tuning is not well established for GABA-INs at eye opening. Over the first week of normal visual experience, excitatory neurons continue to develop orientation selectivity, binocular orientation preference alignment, and direction selectivity[1-4]. During this period, experience-dependent processes drive the development of orientation selectivity, binocular alignment, and direction selectivity for GABA-INs.

In the mouse, it has been hypothesized that the broadening of tuned inhibition gates the plasticity necessary for the monocular matching of orientation preferences[13]. However, the murine visual cortex has 'salt-and-pepper' organization of orientation preferences and low orientation selectivity in interneurons[15,16,22]. Our observations in the modular ferret visual cortex demonstrate that, over a comparable period, when excitatory neurons match orientation preferences[4], GABA-INs are strengthening in orientation selectivity. Thus, our findings challenge the generality of experience-dependent broadening of tuned inhibition in initiating the critical period. Studies of inhibitory development have predominantly focused on the impact of changes in the strength of inhibition, independent of functional properties, as an important factor gating critical period plasticity. Modulating the strength of GABAergic inhibition can alter the timing of the critical period[12,23] and development of modular cortex[24], and transplantation of GABA-INs can rescue critical period-like plasticity[25-27]. Additionally, early dysfunction of GABA-INs leads to impairment of visual circuits[10,11,28]. In contrast, our findings highlight that beyond changes in the strength of inhibition, functional properties of GABA-INs also shift after eye opening, and these changes could contribute to the normal development of the excitatory network.

The influence of GABA-INs on orientation tuning in adult animals has been an area of interest. Early studies suggested that lateral-inhibition shaped orientation tuning by suppressing responses to orthogonal orientations[29,30]. Recent investigations, however, have consistently found weak inhibition for orthogonal orientations- instead, finding untuned or co-tuned inhibition[31-33], suggesting that lateral-inhibition is not required for orientation preferences in mature animals. Yet, there remained the possibility that a transient phase of lateral-inhibition early in development could establish orientation preferences. Our observation that before the onset of sensory experience, GABA-INs in the modular ferret visual cortex are weakly orientation tuned is inconsistent with this possibility. Therefore, orientation preferences in excitatory neurons are likely to result from processes that do not require tuned inhibition.

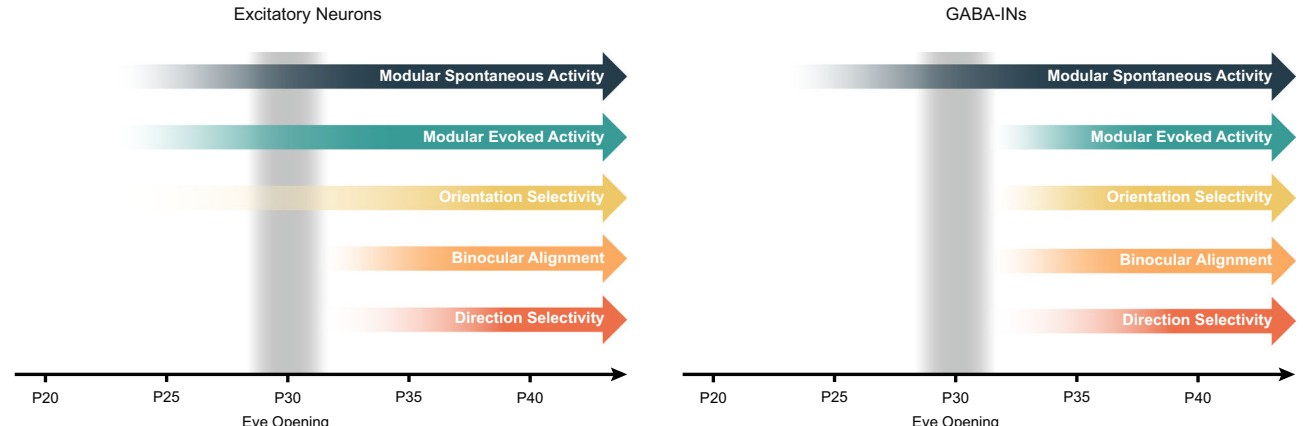

**Fig. 6 Excitatory and Inhibitory neurons undergo distinct development trajectories of functional properties.** Excitatory neurons display modular patterns activity for spontaneous and evoked activity and have initial orientation preferences established before eye opening. Subsequent visual experience aligns binocular and monocular orientation preferences and increases both orientation and direction selectivity. In contrast, GABA-INs only show modular patterns of activity for spontaneous activity before eye opening[45]. Modular evoked activity, orientation preferences and selectivity, binocular alignment, and direction selectivity all develop after eye opening.

GABA-INs encompass a diversity of cells with different genetic markers, cellular morphology, and synaptic targets. For example, interneurons expressing parvalbumin make perisomatic and axonal synapses, while interneurons expressing somatostatin have synapses biased toward the distal dendrites suggesting interneurons subtypes play different roles in shaping cortical activity[34–36]. Additionally, optogenetic modulation of interneuron activity has subtype-specific effects on visually-evoked activity[37–41], and subtype-specific synaptic changes coinciding with eye opening have been observed[42]. Viral expression using the mDlx enhancer labels multiple subtypes of GABA-INs, and differences in functional properties have been observed in mature ferrets[8,17]. Because of biological limitations in identifying cell types early in development[43,44], we pooled all GABA-INs. Importantly, we did not observe significant heterogeneity in cellular responses of Naive animals, suggesting that most GABA-INs do not exhibit modular responses or orientation tuning early in development and that any differences in the developmental progression of GABA-INs labeled by the mDlx enhancer are not reflected in the response to simple stimuli. We are optimistic that novel methods targeting interneurons will allow for the assessment of subtype-specific developmental trajectories of GABA-INs in the future[45].

What contributes to the development of orientation preferences in GABA-INs? We observe that evoked responses show nearly uniform participation of interneurons in naive animals. Thus, it remains likely that a shared visual drive recruits the entire inhibitory network. One possibility is that uniform responses result from lateral inputs from layer 2/3 neurons that have a diversity of orientation preferences. In aggregate, these inputs could drive the non-orientation-specific responses seen early in development, and the subsequent refinement of these inputs could drive the gain of orientation tuning in GABA-INs. Alternatively, untuned responses in interneurons could arise from untuned inputs, possibly originating from the LGN or layer IV that override well-organized lateral inputs from layer 2/3, masking modular GABA-IN activity. Losing these untuned inputs could unmask orientation tuned responses in interneurons. Such a mechanism would be consistent with observations of spontaneous modular inhibitory activity—which likely reflects the structure of intracortical connections[6]—before eye opening[21] and anatomical studies showing the reduction of thalamocortical inputs to GABA-INs across development[46–48]. Investigations into the synaptic inputs to GABA-INs are necessary to clarify these possibilities. In conclusion, our findings highlight interneurons undergo a parallel delayed developmental sequence in which experience-dependent processes drive the maturation of orientation-selective responses, resulting in modular, functional networks of GABA-INs.

## Methods

**Animals**. All experimental procedures were approved by the Max Planck Florida Institute for Neuroscience Institutional Animal Care and Use Committee and were performed in accordance with guidelines from the U.S. National Institute of health. Juvenile female ferrets (*Mustela putorius furo*, Marshall Farms) co-housed with jills on a 16-h light/8-h dark cycle. No a priori sample size estimation was performed, but sample sizes are comparable to other studies which performed in vivo imaging.

**Viral injections and eyelid suturing**. We expressed GCaMP6s by microinjection of a custom-made virus AAV2/1-mDlx-GCaMP6s[31] (Vigene), at 6–21 days before imaging experiments. AAV2/1-mDlx-GCaMP6s was derived from pAAV-mDlx-GCaMP6f-Fishell-2, which was a gift from Gordon Fishell (Addgene plasmid # 83899; http://n2t.net/addgene:83899; RRID:Addgene_83899), and has previously been shown to specifically target a diversity of interneurons subtypes in the ferret visual cortex[8,17]. For excitatory wavelength data, animals were injected with AAV2/1-hSyn-GCaMP6s-WPRE-SV40 (Addgene plasmid # 100843-AAV; http://n2t.net/addgene:100843; RRID:Addgene_100843), which has previously been shown to primarily label excitatory neurons in the ferret visual cortex[8]. Anesthesia induction

was performed using either ketamine (50 mg/kg, IM) and/or isoflurane (1–3%) delivered in $O_2$ and then maintained with isoflurane (1–2%). Atropine (0.2 mg/kg, IM) was administered to reduce secretions, while Buprenorphine (0.01 mg/kg, IM) and a 1:1 mixture of lidocaine and bupivacaine (injected directly into the scalp) were administered as analgesics. Animal temperatures were maintained at 37 °C using a homeothermic heating blanket.

Animals were also mechanically ventilated and both heart rate and end-tidal $CO_2$ were monitored throughout the surgery. Under aseptic surgical technique, a small craniotomy was made over visual cortex 6.5–7 mm lateral and 2 mm anterior to lambda. Approximately 1 μL of virus was pressure infused into the cortex through a pulled glass pipette across two depths (~200 and 400 μm below the surface). For eyelid suturing, eyelids were binocularly sutured during a short aseptic technique between P26-30. Eyelid sutures were monitored daily until they were removed for imaging experiments as noted in the main text.

**Cranial window and acute imaging**. All animals were anesthetized and prepared for surgery as described above. In addition, an IV catheter was placed in the cephalic vein. Preparation for cranial windows and acute imaging was performed as previously described[4]. Briefly, a custom-designed metal headplate (8 mm DIA) was implanted over the injected region and secured with MetaBond (Parkell Inc.). Then a craniotomy and durotomy were performed, exposing the underlying brain. The brain was then stabilized with a custom-designed titanium metal cannula (4.5 mm DIA, 1.5 mm height), adhered to a 4 mm coverslip (Warner Instruments) with optical glue (#71, Norland Products Inc.). Finally, the headplate was hermetically sealed with a stainless-steel retaining ring (5/16" internal retaining ring, McMaster-Carr) and glue (Krazy Glue).

For all imaging experiments, either eyelid sutures were removed, or the eyelids were separated where applicable to ensure visual stimulation was presented to open eyes. Phenylephrin (1.25–5%) and tropicamide (0.5%) were applied to the eyes to retract the nictating membrane and dilate the pupils. Prior to imaging, isoflurane levels were reduced from a surgical plan to ~1–1.5%. After a stable, anesthetic baseline was established for 30 min, animals were paralyzed with pancaronium bromide (0.1–1 mg/kg/h).

**Calcium imaging experiments**. All imaging experiments were performed using a B-Scope microscope (Thorlabs) and began immediately after cranial window surgery. Two-photon experiments of GCaMP6s used a Mai-Tai DeepSee laser (Spectra Physics) at 920 nm for excitation. The B-Scope microscope was controlled by ScanImage (v2015a, Vidreo Technologies) in a resonant-galvo configuration with images acquired at 512 × 512 pixels. Multi-plane images were sequentially collected from one or four imaging planes using a piezoelectric actuator for an effective frame rate of 30 or 6 Hz respectively. Images were acquired at 2× zoom through a 16× water-immersion objective (Nikon, LWD 16X W/0.8 NA) yielding a field-of-view of ~0.7 mm square (1.36 μm/pixel). All two-photon imaging planes were restricted to 100–250 μm from the cortical surface to assess responses in layer 2/3. For widefield epifluorescence imaging experiments, Zyla 5.5 sCMOS camera (Andor) was controlled by μManager (v.1.4.16)[49]. Images were acquired at 15 Hz using a 4 × 4 binning to yield 640 × 540 pixel images through a 4× air objective (Olympus, UPlanFL N 4 ×/0.13 NA).

**Visual stimulation**. Visual stimuli were presented on an LCD screen played ~25 cm from the eyes using PsychoPy (v1.85)[50]. To evoke orientation-specific responses, 0.1 cycles per degree full-field square gratings at 100% contrast drifting at 1 Hz were presented in 16 directions. Low spatial and temporal frequencies were chosen to optimize responses in young animals. In addition, "blank" stimuli of 0% contrast were also presented to establish a threshold for visual responsiveness based on spontaneous activity. All stimuli were randomly interleaved and presented for 4 s followed by 6 s of a gray screen. Timing for visual stimuli and imaging were recorded using Spike2 v7.11b.

**Analysis**. All data analysis was performed using custom-written scripts in either Matlab (v. 2016a or 2018b), Fiji (v1.51n), or Python (Python 3.7), with the following plugins: PIMS v0.4.1, imagio v2.4.1, NumPy v1.15.1, read-roi v1.4.2, scikit-image v0.14.0, SciPy v1.1.0, scikit-learn 1.2.1, and pandas v0.23.4. For both two-photon and widefield epifluorescence imaging data we corrected brain movement during imaging by maximizing phase correlation to a common reference frame. Statistical testing was performed by either bootstrap resampling or Student's paired *t* test, on a per animal basis unless otherwise noted in the text. For bootstrap resampling, 10000 resamplings with replacement were made. Two-tailed assessments of significance were made by comparing observed group averages to the 95th percentile of the extremes (highest 97.5th percentile or lowest 2.5th percentile) of the surrogate distributions. We chose to use bootstrap statistical testing for unpaired data analysis, as it is not dependent on assumptions about the underlying distributions present in our data.

**Two-photon analysis**. To identify cellular ROIs in two-photon imaging, custom software in ImageJ (Cell Magic Wand, v1.0[8]) was used. ΔF/F for each ROI was computed by calculating the $F_0$ using a rolling window (60 s) rank-order filter to the raw fluorescence (20th percentile). Stimulus-evoked responses were computed

by subtracting the mean pre-stimulus ΔF/F (1 s before stimulus onset) from the mean stimulus-evoked response over the entire stimulus period (4 s). Visually responsive cells were defined as cells where the maximum stimulus-evoked response was at least two standard deviations larger than the blank stimulus responses.

Since a fraction of interneurons is expected to have direction-selective responses, all analyses were performed in direction space to minimize any impact of direction selectivity on both cellular and population responses. To assess orientation preference two von Mises functions, constrained to be 180° out of phase, were fit to the trial-median responses to stimulus directions. Orientation preference was assessed as the orientation for the peak response. Cells were considered well-tuned if the $R^2$ of the fit were >0.6. Orientation selectivity was computed using the fit orientation tuning curves:

$$OSI = \frac{r(\theta_{pref}) - r(\theta_{orth})}{r(\theta_{pref}) + r(\theta_{orth})}$$

Where $r(\theta)$ is the fit tuning curve response for either the preferred direction or the orthogonal directions. Similarly, direction selectivity was computed:

$$DSI = \frac{r(\theta_{pref}) - r(\theta_{anti})}{r(\theta_{pref}) + (\theta_{anti})}$$

To test for significant orientation and direction tuning, we computed orientation tuning curve fits, OSI, and DSI for 1000 bootstrap with replacement samplings of trial evoked responses. Cells were deemed significantly orientation or direction tuned if the observed OSI or DSI was larger than the 95th percentile of computed bootstrap OSI or DSI respectively. To assess the variability of cells, trial responses within stimuli were resampled 1000 times to compute resampled tuning curves ($T_n$). The VI was then defined as:

$$VI = 1 - \frac{2!(n-2)!\sum_{i=0}^{n-1}\sum_{j=i+1}^{n}Pearson's\ Correlation\ Coefficient(T_i, T_j)}{n!}$$

We used the Cohen's $d$ metric computed for responses for non-preferred, orthogonal orientation trials and blank trials as the basis for the Stimulus Activity Index (SAI). The SAI allows us to compare non-preferred orientation and blank trials, and to quantify the degree to which observed responses were modulated by the visually driven activity. The SAI was computed using the following formula:

$$SAI = \frac{\mu_{orth} - \mu_{blank}}{\sqrt{\frac{(n_{orth}-1)\sigma_{orth}^2 + (n_{blank}-1)\sigma_{blank}^2}{(n_{orth}+n_{blank}-2)}}}$$

where for both the orthogonal and blank trials: n is the number of trials, μ is the mean and σ is the standard deviation of the trial responses. We choose to use the Cohen's $d$ metric as the basis for our Stimulus Activity Index, as it gives us a normalized difference in activity that takes into account both the average and variance of the responses.

Cellular pairwise correlation maps were generated by computing the Pearson's correlation of cellular responses to square wave drifting grating, independent of the stimulus identity. Short- and long-range cellular correlations were computed by taking the average pairwise correlation for cells within 100 and 300–400 μm respectively. Trial-to-trial pattern correlations were computed by first sorting trials by the direction of the drifting grating, then taking all cells within a field-of-view and computing Pearson's correlation coefficients across trial pairs. Matched and orthogonal trial-to-trial correlations were computed as the mean correlation for matched and orthogonal stimuli. We chose to perform these comparisons in direction space to minimize differences in evoked patterns due to increasing direction selectivity across development.

Finally, we implemented a normalized template matching algorithm as previously described[4]. Briefly, template population responses for each stimulus condition were generated by computing the median response using half the trials across all direction conditions. We assessed the performance of the template decoder by predicting the presented stimuli for the trials not used to generate the template population responses and altered the number of cells included in the templates to generate template decoding accuracy versus template size. To assess the overall performance of the template decoders, we chose to use 40 cell templates, as the accuracy across all conditions appeared to approach peak accuracy near template size. To assess chance decoding rates, we generated templates using shuffled trials and assessed the decoding accuracy of a shuffled test data set ($n = 1000$).

**Widefield epifluorescence analysis**. ROI segmentation was performed in widefield epifluorescence imaging by manually drawing around cortical regions where robust visual evoked activity was observed. For analysis, all images were spatially downsampled by a factor of 2× to yield 320 × 270 pixels at a spatial resolution of 11.63 μm/pixel. Slow drifts in fluorescence intensity were eliminated by calculating the ΔF/F. For each pixel, the baseline F0 was calculated by applying a rank-order filter to the raw fluorescence trace (10th percentile) with a rolling time window of 60 s. Responses were filtered with a previously published spatial bandpass filter[4] with the low-pass cutoff defined as 50 μm and the high-pass filter cutoff as

3200 μm. Preferred orientation was computed as the vector sum of the average response for each orientation:

$$z = \sum_k R(\theta_k)\exp(2i\theta_k)$$

$$\theta_{pref} = \tan^{-1}\frac{real(z)}{imaginary(z)}$$

where $R(\theta)$ is the median response for a given orientation θ. To evaluate orientation-selective pixels, we computed $1 - CV$ metrics using the following formula:

$$1 - CV = \left|\frac{\sum_k R(\theta_k)\exp(2i\theta_k)}{\sum_k R(\theta_k)}\right|$$

Significantly tuned pixels were assessed by computing $1 - CV$ for 1000 bootstrapped with replacement resamplings. Pixels were deemed significant if the observed $1 - CV$ was greater than 95% of the resampled $1 - CV$s. Correlation maps were computed using seed pixels and calculating the Pearson's correlation coefficient for all other pixels with the ROI. Correlations were then binned by pixel distances to generate correlation as a function of distance plots, and linearity ($R^2$) was assessed by performing ordinary least squares linear progression. Finally, trial-to-trial correlation matrices were generated by computing the Pearson's correlation coefficient for all possible combinations of trials. Trial-to-trial correlations for matched and orthogonal stimuli were computed by taking the average correlation for sets of trials with matched stimuli and orthogonal stimuli in direction space.

Wavelength analysis was adapted from a previously established method for determining the spacing of columns in imaging experiments[19]. Briefly pairwise pixel correlation maps (for a grid of points spaced by 150 nm spanning the ROI) were fit by Morlet wavelets. We chose to use the pairwise pixel correlation maps to enhance the modular structure that underlies all visually-evoked responses. Additionally, we fit Morlet wavelets for a uniform correlation across the ROI as a control. Relatively large, isotropic wavelets ($k = 7$ and =1) were used to measure the wavelength of the correlation-based columns. Each wavelet rotated between 24 uniformly spaced orientations and the characteristic wavelengths were also varied from 0.5 to 1 mm with a step size of 0.005 mm. Wavelengths were chosen as the mean characteristic wavelength of the wavelet that optimally fit the pairwise pixel correlation maps for each pixel within the ROI.

**Statistics**. Our cellular analyses were based on 781 cells from 7 Naive animals, 1185 cells from 6 Brief experience animals, 541 cells from 5 Extended experience animals, 394 cells from 4 Binocular Deprivation animals, and 310 cells from 4 Recovery animals. Widefield calcium imaging was based on 7 Naive, 6 Brief experience, 5 Extended experience, 4 Binocular Deprivation, and 4 Recovery animals. Wavelength analysis for excitatory neurons was based on 5 Naive, 4 Brief experience, 5 Extended experience, 4 Binocular Deprivation, and 5 Recovery animals. Some of the animals used for excitatory analysis have been included in a previous study[4]. Unless otherwise noted, hypothesis testing was performed using a bootstrapping test on a per animal basis. We deliberately chose to use resampling methods for hypothesis testing, rather than relying on standard parametric or non-parametric tests, to avoid confounds that could arise from making assumptions on the underlying sampling distributions present in our data. Statistical significance was assessed by ranking if the observed difference between group averages was more extreme than the 95th percentile from a surrogate distribution, where the null hypothesis is true. All statistical tests were two-tail, except when assessing if the performance of the template matching decoder was above chance levels (one-tail). For each hypothesis test, we produced a relevant surrogate distribution by pooling data between groups, bootstrapped new group averages from the pooled data using replacement ($n = 1000$), and then computed a distribution of differences between surrogate group averages. Paired tests for pattern correlation and response clustering were performed using a two-tailed paired Student's $t$ test.

**Reporting summary**. Further information on research design is available in the Nature Research Reporting Summary linked to this article.

## Data availability
Raw data are available from the corresponding author upon reasonable request, because data are to large to share in a public depository. Source data are provided with this paper.

## Code availability
All analysis code for this study can be found at https://github.com/jtchang/ChangAndFitzpatrick2022. Code are also available from the corresponding author upon reasonable request.

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

## Acknowledgements

We would like to thank N. Shultz, R. Satterfield, J. Kerr, G. Kreal, and D. Whitney for technical assistance, as well as members of the Fitzpatrick laboratory for helpful discussions. This research was supported by US National Institutes of health grants EY011488 (DF) and EY026273 (DF) and the Max Planck Florida Institute for Neuroscience.

## Author contributions

All authors (J.C and D.F.) designed the study, analyzed the results, and wrote the paper. J.C. performed the widefield and two-photon calcium imaging.

## Funding

## Competing interests

The authors declare no competing interests.
