## [Peer Review File · Nature Communications]

Development of visual response selectivity in cortical GABAergic interneuronsREVIEWER COMMENTS

Reviewer #1 (Remarks to the Author):

In their study, Chang and Fitzpatrick analyze the development of visual responses in ferret visual cortex interneurons. Using a viral promoter (Dlx) that preferentially targets GABAergic interneurons (GABA-INS) to express the genetically-encoded calcium indicator GCaMP6s, the authors examine the orientation tuning of individual GABA-INS by two-photon imaging and their modular population organization by wide-field epifluorescence and two-photon imaging. The authors report that individual GABA-INS, unlike excitatory neurons, are poorly tuned to stimulus orientation at eye-opening but become orientation-selective shortly after that. This increase in orientation tuning requires visual experience. The window of experience-dependent plasticity appears to be relatively wide, and orientation tuning reaches normal levels if visual experience is introduced at postnatal day 36 (P36). Similarly, the modular organization of GABA-INS' orientation tuning and activity correlations appear to develop after eye-opening in an experience-dependent manner.

Overall, this is an interesting study, revealing a delayed maturation of GABA-INS and their modular organization compared to excitatory neurons and suggesting activity-dependent mechanisms may drive this maturation. However, the authors need to clarify, experimentally, whether the observed changes in apparent tuning and modular organization are due to changes in light responses or spontaneous activity. In addition, they should show whether the delayed maturation applies equally to diverse GABA-INS.

Specific comments

1.) The apparently less refined tuning of GABA-INS to stimulus orientation could either reflect GABA-INS responding to more diverse visual stimuli (i.e., truly less refined tuning) or contamination by spontaneous activity that is not entrained to the visual stimulus. The authors should record the spontaneous activity of GABA-INS across age and test whether a combination of tuned responses plus the spontaneous activity of the observed frequency could account for the apparently less refined GABA-INS tuning in 'naive' ferrets.

2.) In addition, the authors could analyze the temporal jitter of apparent responses to try to evaluate which of them are likely stimulus-driven and which are likely coincidental (i.e., spontaneous events occurring during the stimulus window). They could then try to subtract coincidental responses to explore the underlying visual tuning.

3.) Similar considerations apply to the modular organization analyzed by epifluorescence and two-photon imaging. It seems that around eye-opening activity correlations of nearby neurons are uniformly high. Later, patterns begin to vary locally in a manner that aligns with the cells' orientation preferences. The authors should test whether the same changes are seen in the spontaneous activity and whether, given the observed frequency of events, it could account for the overall observed organization (or quantify its fractional contribution).

4.) If spontaneous activity contributes to the observed changes in tuning, the authors should test whether sensory deprivation increases/prolongs its predominance?

5.) If spontaneous activity contributes to the observed changes in tuning, the authors should compare spontaneous activity between GABA-INs and excitatory neurons to see if it can account for/if it contributes to the apparent delay in the visual maturation of the former.

6.) GABA-INs are a diverse class of cells. The authors should test whether their main observation (i.e., the delay in the visual maturation and persistence of correlated spontaneous activity) applies to the subclasses they previously distinguished by post hoc immunohistochemistry (i.e., Gad+ only, SOM+, and PV+) (Wilson et al. Neuron 2017).

Reviewer #2 (Remarks to the Author):

The authors examined developmental changes of visual responses of interneurons (INs) after eye-opening in the ferret V1 and found that orientation selectivity is poor at eye-opening and matured several days after eye-opening depending on the visual experience. At the population level, widespread activity was observed at eye-opening. The widespread activity reduced independent of the visual experience and clustered activity emerged after several days of visual experience, suggesting the visual experience-dependent and independent mechanisms of the development. Together with the previous study about excitatory cells, the authors suggest the delayed maturation of the inhibitory network.

Although the argument of delayed maturation of inhibitory neurons has already been reported in the mouse V1 (Kulman, Trachtenberg et al., 2011), investigation in the animal with functional columns is a very important theme and attractive for many readers in the sensory neuroscience field. The authors further reveal the developmental change of population activity pattern in INs which has not been previously reported. However, I feel that there are several concerns that should be addressed before the publication. Additional data and analyses would improve the manuscript.

Major points

1. The authors reported widespread, visually evoked activity in naïve animals. I feel that the descriptions about the widespread activity is not enough to understand what kinds of activity was actually observed in naïve animals. Please more clearly describe activity patterns in naïve animals.

First, I wonder whether this activity is visually evoked or reflects spontaneous activity which occurs during visual stimuli. For example, analyzing the reliability of onset timing should be useful to reveal this point.

Second, please demonstrate more examples of activity patterns (e.g., raster plot shown in Fig2a, activity pattern map in supplementary figure 2b, a movie of the widespread activity, etc.).

Third, it is not clear how often the widespread visually evoked activity is observed. For example, in the activity pattern of naïve animal shown in the supplementary figure 2b, population response patterns highly varied across trials and the pattern looks clustered in some trials: widespread activity across all cells is observed in the trial 1 and 3, some clustered activity in trial 4, and almost no activity in trial 2 and 5. According to these results, pattern correlation would not be so high in naïve animals. However, the pattern correlation matrix shows very high values across trials/stimuli (figure 2b), suggesting the pattern of widespread activity is consistent across trials. I am puzzled by this inconsistency. When the authors calculate the pattern correlation (figure 3b), the average activation across cells is subtracted (by definition of the correlation), which might underestimate the amplitude variability. Analyses of population-averaged activity and participation rate may be informative.

Fourth, related to the above point, I am interested in how often the clustered patterns are observed in naïve animals. If the clustered activity is observed with a substantial frequency in naïve animals, it may be a seed of the clustered pattern observed in a later stage of development. Is the clustered activity pattern variable across trials? Another possibility is that clustered activity is already established in naïve animals, but it was masked by a widespread activity. If the widespread pattern is removed and pattern correlation is computed for the remaining data, is the resulting pattern correlation matrix similar to that in observed in the matured animals?

2. I am curious about the spontaneous activity of INs at eye-opening, its developmental change, and its relationship to the visually evoked activity. The authors' group has previously reported that spontaneous activity of mainly excitatory neurons (GCaMP expressing cells under synapsin promotor) is spatially clustered and related to the functional column at eye-opening (Smith et al., 2018). Thus, it should be important to examine whether the spontaneous activity of INs at eye-opening is also widespread as the visually evoked activity or spatially clustered as the excitatory cells. Revealing the similarity/dissimilarity between spontaneous and sensory-evoked activity should provide an important clue to speculate underlying mechanisms. In addition, developmental change of IN's spontaneous activity and its comparison to that of excitatory cells is an important theme and it must increase the impact of this manuscript.

3. I think that the difference in correlations for short- and long-range cell pairs (figure 3g and 5f) is not a good measure of modularity. It just shows the existence of some clustering but does not show the modular periodical pattern of functional maps. Please consider other measures of modularity.

4. Based on the results of binocular deprivation (BD), the authors argue that the development of orientation tuning depends on the visual experience, whereas the reduction in the widespread responses is independent of the visual experience. “the difference in correlations for short- and long-range cell pairs was similar when compared to animals with normal extended experience (BD 0.12 ± 0.03 vs Extended 0.19 ± 0.02 , $p=0.0630$, $n= 4$ vs 4 , Mean \pm SEM, bootstrap test).” However, the difference in correlations decreases from 0.19 to 0.12 (see also fig 5f and fig 3g), although the difference is not marginally significant ($p=0.06$). I wonder whether the reduction in the widespread activity in BD animals is really the same as the reduction in the naturally reared animals and whether the reduction in the widespread responses is completely independent of the visual experience.

Further, as pointed out in 3), I do not think that the difference in correlations for short- and long-range cell pairs is not a good measure of modularity. Only with this index, I do not think the modular pattern in the BD animal is firmly supported. Please demonstrate the correlation maps in BD animals in figure 5.

5. The authors examined the difference in the monocular orientation preference mismatch between Brief and Extended, but not Naïve. Although the percentage of the orientation-selective cells in naïve animals is smaller (around 25%), it should be possible to examine the monocular orientation preference mismatch in these cells. The authors suggest that “inhibitory and excitatory neurons have fundamentally different developmental progressions for orientation selectivity and binocularly aligned responses”, but it is strange to compare naïve and brief for excitatory neurons and brief and extended for inhibitory neurons. In order to suggest that the binocularly aligned responses are also different in inhibitory neurons, it is necessary to analyze the binocular mismatch in inhibitory neurons in naïve animals

6. The clustered activity in the extended group was relatively weak compared to the brief group (e.g., Figure 3b-g), and the authors speculate this could result from the diversification of neuronal preference and/or cortical expansion (L216-217). Another possibility is an artifact resulting from the reduction of visual responses due to excessive expression of GCaMP. Please exclude this possibility.

Minor point

The authors only demonstrate the data of INs in the manuscript. As commented above, the authors' group has reported the data of (mainly) excitatory cells. Direct comparison between excitatory and inhibitory cell is informative and improve the impact of this manuscript. The schematic diagram which directly compares the result of INs to those of excitatory cells in literature is very useful for readers who are not familiar with the topics to recognize the results of this manuscript.

In figure 3b and c, the map area in a naïve animal is significantly smaller than other groups. Why is the area in naïve animals small? In figure 3e, pixel-pixel distance is plotted up to 1.5 mm. Thus, there should be some data in naïve animals which cover more than 1.5 mm. The small size of the correlation map in figure 3c prevents the readers to evaluate the modular structures. I suggest demonstrating the wider maps in naïve animals.

The monocular orientation mismatches in inhibitory neurons in Brief animals (8.54 degree) is smaller than those in excitatory neurons (12.73 degrees). Does the author have any suggestion about why the mismatch is smaller in inhibitory neurons?

I could not find how the authors calculated the pattern correlation matrix. The description in the method is very sketchy. The trial number of the pattern correlation matrix seems to correspond to the direction of stimuli because adjacent trials tend to show high correlations. Please clearly describe this point in the legend.

I could not find the description of the spatial frequency in the section of Visual stimulation (L464). Please describe it. Is there any possibility that poor visual response in naïve animals is due to non-preferred spatial and/or temporal frequency of grating stimuli? Please exclude this possibility.

In figure 3c, the seed point should have a correlation value of 1. However, in the extended animals, the value seems smaller than 1 (although the seed point itself is hidden). Why?

In figure 4, data of the brief group should be plotted, and a comparison between BD and the brief group should be demonstrated in the figure, although they presented the results in the manuscript.

In the template matching analysis, please clarify that decoder performance was tested with data from the trials that were not used to make the templates.

“Age matched animals with normal visual experience also displayed higher similarity of response patterns for matched stimuli, and lower similarity for orthogonal stimuli (matched Recovery 0.81 ± 0.02 vs Extended 0.76 ± 0.02 , $p = 273 \ 0.1118$; orthogonal Recovery 0.25 ± 0.02 vs Extended 0.09 ± 0.02 , $p = 0.0076$; $n = 5$ vs 4 , Mean \pm SEM, 274 bootstrap test).”

“lower similarity” should be “higher similarity”.

L107. Reference number 15 seems to be wrong.

L672. Reference number should be 45.

L583. Chang, J. T., Whitney, D. & Fitzpatrick, D. Experience-Dependent Reorganization Drives Development of a Binocularly Unified Cortical Representation of Orientation. *Neuron* 0, (2020). Please add the volume number and page number.

We thank the reviewers for their comments regarding our manuscript. We have made a number of changes in response to reviewer concerns. Changes are outlined with the reviewer comments (bold text) below. All new text changes are highlighted as red font in the manuscript and excerpts are duplicated in some of the responses below.

Reviewer #1 (Remarks to the Author):

In their study, Chang and Fitzpatrick analyze the development of visual responses in ferret visual cortex interneurons. Using a viral promoter (Dlx) that preferentially targets GABAergic interneurons (GABA-INs) to express the genetically-encoded calcium indicator GCaMP6s, the authors examine the orientation tuning of individual GABA-INs by two-photon imaging and their modular population organization by wide-field epifluorescence and two-photon imaging. The authors report that individual GABA-INs, unlike excitatory neurons, are poorly tuned to stimulus orientation at eye-opening but become orientation-selective shortly after that. This increase in orientation tuning requires visual experience. The window of experience-dependent plasticity appears to be relatively wide, and orientation tuning reaches normal levels if visual experience is introduced at postnatal day 36 (P36). Similarly, the modular organization of GABA-INs' orientation tuning and activity correlations appear to develop after eye-opening in an experience-dependent manner.

Overall, this is an interesting study, revealing a delayed maturation of GABA-INs and their modular organization compared to excitatory neurons and suggesting activity-dependent mechanisms may drive this maturation. However, the authors need to clarify, experimentally, whether the observed changes in apparent tuning and modular organization are due to changes in light responses or spontaneous activity. In addition, they should show whether the delayed maturation applies equally to diverse GABA-INs.

We thank the reviewer for these comments. We hope that the reviewer's concerns are addressed by the additional analyses and clarifications we have added to our manuscript and outline below.

Specific comments

1.) The apparently less refined tuning of GABA-INs to stimulus orientation could either reflect GABA-INs responding to more diverse visual stimuli (i.e., truly less refined tuning) or contamination by spontaneous activity that is not entrained to the visual stimulus. The authors should record the spontaneous activity of GABA-INs across age and test whether a combination of tuned responses plus the spontaneous activity of the observed frequency could account for the apparently less refined GABA-IN tuning in 'naive' ferrets.

Indeed, the untuned GABA-IN responses could be due to responses to diverse visual stimuli or contamination by spontaneous activity. In our analyses, we accounted for the contribution of spontaneous activity to our measured results in the method in which we computed the responses to visual stimulation. As outlined in the Methods section of our manuscript, we first used a 60-second rolling window to compute the baseline fluorescence (F_0) and $\Delta F/F$. This window allows us to capture slow changes in fluorescence over time such as those that could be attributed to photobleaching. To control for shorter time scale events, such as spontaneous activity, our stimulus responses were

computed as the difference in $\Delta F/F$ for the stimulus period and the pre-stimulus period (1 second period before stimulus onset). This difference allows us to discount activity that cannot be attributed to stimulus presentation. In addition, all metrics presented in our manuscript use median responses for each orientation, which minimizes the contribution of outlier trials that could be related to spontaneous activity. Thus, our analyses were chosen to minimize the impact of spontaneous activity on the measured activity during visual stimulation.

To further control for the possibility of spontaneous activity contaminating our measures of visually driven responses, we presented blank trials (trials without a visual stimulus) and characterized the proportion of GABA-INS that had responses to visual stimuli that exceeded two standard deviations of the values on blank trials. This threshold allows us to identify neurons that are active due to visual stimulation and excludes neurons whose activity cannot be distinguished from spontaneous activity by our analyses. As shown in the example for Fig. 1c, almost all GABA-INS in Naive animals had evident visually evoked activity that was larger than that observed in blank trials. Consistent with these observations, an overwhelming majority of GABA-INS were responsive in Naive animals (fraction of responsive cells, Naive 0.998 ± 0.001). Furthermore, when we subtract the average fluorescence observed during blank trials from visual responses, we do not observe a significant shift in the proportion of significantly orientation or direction selective GABA-INS in Naive animals (fraction of significantly orientation selective cells, uncorrected 0.24 ± 0.09 vs corrected 0.24 ± 0.09 , $p = 0.8605$; fraction of significantly direction selective cells, uncorrected 0.10 ± 0.03 vs corrected 0.08 ± 0.02 , $p = 0.0839$; $n = 7$, Mean \pm SEM, fraction Student's paired t-test). Thus, measured visual responses for GABA-INS are not consistent with activity that is dominated by spontaneous activity in GABA-INS.

While the analyses in our study are restricted to visually responsive GABA-INS, we thought it was important to specifically address the degree to which the poor tuning of these visually responsive neurons in Naive animals was attributable to visual drive; i.e. that these neurons exhibit visual responses to a broad range of orientations. To explicitly test the contribution of visually driven responses to weak orientation selectivity we have expanded our analyses to compare cellular activity during blank trials and cellular activity in response to non-preferred (orthogonal) orientations. To compare the activity, we computed the Stimulus Activity Index (SAI), which uses the Cohen's d metric to quantify the difference between blank trials and non-preferred trials (see Methods). SAI gives us a quantitative measure of the degree to which activity for presentations of the non-preferred orientation deviates from that expected for spontaneous activity. If visual drive is the predominant contributor to the increased activity associated with presentation of the non-preferred stimulus orientation, we would expect positive SAI values.

In Naive animals, significantly orientation selective GABA-INS show a positive modulation (SAI significantly larger than zero) for visual stimulation with the non-preferred orientation (Supplementary Fig. 1a, SAI Naive 1.43 ± 0.05 , $p < 0.0001$, $n = 233$; Student's t-test). This positive modulation also held for cells in Brief and Extended experience animals (Brief 1.00 ± 0.03 , $p < 0.0001$, $n = 966$; Extended 0.17 ± 0.04 , $p < 0.0001$, $n = 326$; Student's t-test). Interestingly, we observed that the degree to which visual stimulation drives activity to the non-preferred orientation actually decreases with experience (SAI Naive 1.43 ± 0.05 vs Brief 1.00 ± 0.03 , $p < 0.0001$, $n = 233$ vs 966 ; Naive 1.43 ± 0.05 vs Extended 0.17 ± 0.04 , $p < 0.0001$, $n = 233$ vs 326 ; Mean \pm SEM, bootstrap test), indicating that non-preferred responses are more similar to spontaneous activity in older animals. This suggests that across all ages there is a positive modulation for visual stimulation in the non-preferred orientation and that the reduction in

visual drive to the non-preferred orientation could be one factor contributing to the strengthening of GABA-IN orientation selectivity following the onset of visual experience.

Finally, if visually driven responses to the non-preferred orientation actively contribute to the poor orientation selectivity evident in the GABA-INs of Naive animals, we would expect to see some evidence for this in the relationship between a neuron's OSI and SAI. Neurons with weak orientation selectivity (lower OSI) would be expected to show higher levels of visually-evoked activity for the non-preferred orientation (higher SAI) than those with stronger orientation selectivity; i.e. a negative correlation. To address this, we examined the correlation between orientation selectivity (OSI) and the SAI for GABA-INs that displayed significant orientation selectivity (Supplementary Fig. 1b).

Consistent with this hypothesis, we observe that GABA-INs in the Naive age group, had a weak, but significant negative correlation between SAI and OSI (Naive $r = -0.28$, $p < 0.0001$, $n=233$, Pearson's r). Therefore, in Naive animals, GABA-INs have weak OSI that can be attributed to "truly less refined tuning" as GABA-INs are responding strongly to more orientations.

In sum, our analyses demonstrate that spontaneous activity alone is insufficient to account for the weak orientation tuning of GABA-INs in early development. Instead, visual responses to the non-preferred orientations contributes to the observed weak orientation selectivity in GABA-INs early in development. We have updated our text to reflect these new analyses, highlighting the role of visually evoked activity for the orthogonal orientation as a contributor to weak orientation selectivity of GABA-INs early in development. Updated text has been reproduced below (lines 98-130):

Does weak orientation tuning arise from visually evoked activity? Were lack of visual responsiveness or a high degree of spontaneous activity to play a major role in the poor orientation tuning, we would expect to observe fewer visually responsive neurons early in development. We do not observe this. We presented blank trials (trials without a visual stimulus) and characterized the proportion of GABA-INs that had responses to visual stimuli that were larger than two standard deviations as compared to blank trials. Most GABA-INs were visually responsive, and this fraction did not significantly change (Fig. 1i, Fraction of responsive cells, Naive 0.998 ± 0.001 vs Brief 0.995 ± 0.003 , $p=0.3412$, $n= 7$ vs 6 ; Naive 0.998 ± 0.001 vs Extended 0.975 ± 0.013 , $p=0.0647$, $n= 7$ vs 5 ; Mean \pm SEM, bootstrap test). These data suggest that activity of GABA-INs observed during stimulus presentation reflect visually driven activity for all age groups.

While peak cellular responses reflect visually evoked activity, one question that may arise is whether activity to the non-preferred orientation also reflected visually evoked activity. To test this, we computed a Stimulus Activity Index (SAI, see Methods) by comparing responses to the non-preferred orientation and blank trials for cells that were significantly orientation-selective. The SAI reflects the degree to which observed cellular responses deviate from the predicted response due to spontaneous activity. Positive SAIs reflect visually-driven activity, negative SAIs reflect visually-suppressed activity, and an SAI of zero reflects activity indistinguishable from spontaneous activity. For all age groups, GABA-INs had SAIs significantly larger than zero, (Supplementary Fig. 1a, SAI Naive 1.43 ± 0.05 , $p < 0.0001$, $n=233$ cells; Brief 1.00 ± 0.03 , $p < 0.0001$, $n=966$ cells; Extended 0.17 ± 0.04 , $p < 0.0001$, $n=326$ cells; Mean \pm SEM, Students' t -test). Thus, visual stimulation increases the activity of GABA-INs even when the non-preferred orientation is presented.

We also observed that the degree to which visual stimulation drives activity to the non-preferred orientation decreased with experience (SAI Naive 1.43 ± 0.05 vs Brief 1.00 ± 0.03 , $p <$

0.0001, n=233 vs 966 cells; Naive 1.43 ± 0.05 vs Extended 0.17 ± 0.04 , $p < 0.0001$, $n = 233$ vs 326 cells; Mean \pm SEM, bootstrap test). This raises the possibility that visually evoked activity to the non-preferred orientation contributes to the weak orientation selectivity of GABA-INS early in development. If this is the case, we would expect to see an inverse relationship between OSI and SAI- as the visually evoked response to the non-preferred orientation weakens (lower SAI), orientation selectivity increases (higher OSI). Indeed, GABA-INS displayed a weak but significant negative correlation between OSI and SAI in Naive and Brief, but not Extended experience animals (Supplementary Fig. 1b, Naive $r=-0.28$, $p < 0.0001$, $n=233$ cells; Brief $r=-0.25$, $p < 0.0001$, $n=966$ cells; Extended $r= -0.05$, $p=0.3730$, $n=326$ cells; Pearson's r). Together these findings suggest visual responsiveness to the non-preferred orientation is a key factor lowering orientation selectivity of GABA-INS early in development.

2.) In addition, the authors could analyze the temporal jitter of apparent responses to try to evaluate which of them are likely stimulus-driven and which are likely coincidental (i.e., spontaneous events occurring during the stimulus window). They could then try to subtract coincidental responses to explore the underlying visual tuning.

Indeed, the entrainment of responses to visually evoked activity (visual responsiveness), can often be characterized by the temporal jitter of apparent responses. The slow kinetics of GCaMP6s¹ and the movement of the drifting square gratings used as visual stimuli, however, preclude us from using this criterion for evaluating visual responsiveness. We believe, however, that the analyses outlined above have sufficiently implicated visual responsiveness to non-preferred orientations as a contributor to the weak orientation tuning observed.

3.) Similar considerations apply to the modular organization analyzed by epifluorescence and two-photon imaging. It seems that around eye-opening activity correlations of nearby neurons are uniformly high. Later, patterns begin to vary locally in a manner that aligns with the cells' orientation preferences. The authors should test whether the same changes are seen in the spontaneous activity and whether, given the observed frequency of events, it could account for the overall observed organization (or quantify its fractional contribution).

The organization of spontaneous activity is an area of interest, and one that our lab has previously investigated in excitatory networks. Work by *Mulholland et al. 2019* demonstrated that spontaneous activity for GABA-INS displays modular patterns in ferrets before eye opening. Importantly, modular spontaneous activity fails to account for the uniform pattern of correlations we observed in animals before eye opening. Furthermore, as previously discussed, we ruled out stimulus-independent spontaneous activity as the sole contributor to GABA-IN activity during stimulus presentation. In sum, these findings highlight that lack of modular organization before eye opening is observed for visually evoked responses and cannot be attributed to intrinsically generated patterns of activity.

Previously we had included a reference to *Mulholland et al. 2019* in our discussion, and as these recent findings of modular GABA-IN activity before eye opening have recently been published in *eLife*, we have cited this work in our new discussion paragraph (see Reviewer #2 Minor Comments), and also updated the citation (21; lines 445-461).

*What contributes to the development of orientation preferences in GABA-INs? We observe that evoked responses show nearly uniform participation of interneurons in naive animals in primary visual cortex. Thus, it remains likely that a shared visual drive recruits the entire inhibitory network. One possibility is that uniform responses result from lateral inputs from neighboring layer II/III neurons that have a diverse spectrum of orientation preferences. In aggregate, these inputs could drive the non-orientation-specific responses seen early in development, and the subsequent refinement of the orientation preferences of these inputs could drive the gain of orientation tuning in interneurons. Alternatively, untuned responses in interneurons could arise from untuned inputs, possibly originating from the LGN or layer IV that **override** well-organized lateral inputs from layer II/III during visual stimulation, **possibly masking modular GABA-IN network activity**. The loss of these untuned inputs could unmask orientation tuned responses in interneurons after eye opening. Such a mechanism would be consistent with recent observations of spontaneous modular inhibitory activity before the onset of vision²¹, and anatomical studies that have established reduction in thalamocortical inputs to cortical interneurons across early development⁴⁶⁻⁴⁸. Further investigation into the synaptic inputs to GABA-INs is necessary to clarify these possibilities. In conclusion, our findings highlight that interneurons undergo a parallel, delayed **developmental sequence** in which experience-dependent processes drive the **maturation** of orientation-selective responses, resulting in modular, functional networks of GABA-INs.*

4.) If spontaneous activity contributes to the observed changes in tuning, the authors should test whether sensory deprivation increases/prolongs its predominance?

As we have shown above, responses, even to the non-preferred orientation (as measured by SAI), are positively modulated by visual stimulation. Thus, we do not believe that spontaneous activity dominates the observed responses nor contributes to weak GABA-IN orientation tuning observed in Naive animals. Although further experiments are required to fully answer the question if delayed onset of visual experience alters spontaneous activity, neither an increase nor a decrease in the frequency of spontaneous activity would fundamentally alter the conclusions we have reached.

Nevertheless, we examined the responses for GABA-INs in animals after binocular deprivation to again rule out the contribution of spontaneous activity as a contributing factor to weak orientation selectivity in BD animals. As we observed with Naive, Brief and Extended experience animals, tuned GABA-INs in BD animals showed activity to the orthogonal orientation that was significantly higher than would have been predicted for spontaneous activity as measured by the SAI (Reviewer Fig. 1a; BD 1.16 ± 0.06 , $p < 0.0001$, $n=102$; Student's t-test). These findings suggests that spontaneous activity alone cannot account for responses to the non-preferred orientation in BD animals.

Next, we examined the relationship between OSI and SAI. As we have shown above (Reviewer 1 comment #1), weak orientation selectivity is negatively correlated with the response to the orthogonal orientation for GABA-INs in Naive animals- suggesting that weak orientation selectivity of GABA-INs can in part be attributed to responsiveness to non-preferred orientations. In contrast, for BD animals we observed a weak negative correlation, which failed to reach significance (Reviewer Fig. 1b; $r=-0.10$, $p=.3038$, $n=291$ cells). Weak orientation selectivity in BD animals thus likely arises from other factors which may play less of a role in the weak orientation selectivity of GABA-INs in Naive animals.

These observed differences between Naive and sensory deprived animals are consistent with findings in excitatory neurons, where delaying the onset of visual experience by eyelid suture does not merely delay the development of visual cortex²⁻⁴. We also observed differences between Naive and BD animals in the impaired short-range clustering and reduced long-range correlations (Fig. 5f). In sum, spontaneous activity does not account for weak orientation tuning in Naive or BD animals, however, the factors that lead to weak orientation tuning in BD animals are likely different than those in Naive animals.

Reviewer Fig. 1 Visual responsiveness to the non-preferred orientation for animals after the delayed onset of visual experience. (a) Violin plot and Mean \pm SD of Orthogonal-blank Cohen's d for significantly orientation-selective GABA-INs in BD animals. ***: $p < 0.005$, Student's t -test. (b) Scatter plots of the discriminability of the orthogonal orientation and blank trials (Orthogonal-Blank Cohen's d) versus orientation selectivity index (OSI) for BD animals. Cells shown are significantly selective for orientation.

5.) If spontaneous activity contributes to the observed changes in tuning, the authors should compare spontaneous activity between GABA-INs and excitatory neurons to see if it can account for/if it contributes to the apparent delay in the visual maturation of the former.

As we have shown above, responses, even to the non-preferred orientation, are positively modulated by visual stimulation, thus we do not believe that spontaneous activity dominates the observed responses nor contributes to changes in the orientation tuning of cells. Furthermore, as we have previously discussed, *Mulholland et al.* demonstrate that even early in development, excitatory and inhibitory neurons have tightly-coupled spontaneous activity⁵ (see our discussion for Reviewer 1 Comment #3). These observations of tightly-coupled excitatory and inhibitory spontaneous activity would predict that GABA-INs before eye opening should display orientation tuned responses similar to those observed for excitatory neurons. Instead, we observe weak orientation tuning, and a low fraction of significantly tuned GABA-INs before eye opening. Thus, spontaneous activity and the coupling of excitatory neurons and GABA-INs during spontaneous activity fail to account for the strengthening of orientation tuning after eye opening for GABA-INs.

6.) GABA-INs are a diverse class of cells. The authors should test whether their main observation (i.e., the delay in the visual maturation and persistence of correlated spontaneous activity) applies to the subclasses they previously distinguished by post hoc immunohistochemistry (i.e., Gad+ only, SOM+,

and PV+) (Wilson et al. Neuron 2017).

Understanding whether the trajectory of development varies across subclasses of GABA-INs is certainly an area of interest that we would like to pursue in the future. As the reviewer points out, our lab has already made some progress at establishing differences in the properties of neurons based on molecular markers such as somatostatin and parvalbumin in mature animals. However, as we have outlined in our discussion and as demonstrated by other groups, assessing these subclasses in the developing neocortex is difficult due to low levels of expression of some of these markers early in development⁶. Importantly, we do not think that identifying these markers will markedly alter our present findings, as we did not observe groups of neurons that did not participate in network wide activity during visual stimulation before eye opening.

We have made updates to our discussion section further emphasize that the biological limitations in identifying GABA-INs subclasses in early development, and how the classification of GABA-IN subclasses would not alter our interpretation about how the inhibitory network responds to visual stimuli before eye opening. We have reproduced the relevant discussion paragraph below for your convenience (lines 427-444):

*GABA-INs encompass a diversity of cells with different genetic markers, cellular morphology, and synaptic targets in the visual cortex. For example, interneurons expressing parvalbumin make perisomatic and axonal contacts, while interneurons expressing somatostatin have synaptic contacts biased toward the distal dendrites suggesting subtypes of interneurons may play different roles in shaping cortical activity³⁶⁻³⁸. Additionally, optogenetic modulation of interneuron activity has subtype-specific effects on visually evoked activity³⁹⁻⁴³, and subtype-specific synaptic changes coinciding with the onset of visual experience have also been observed⁴⁴. Viral expression using the mDlx enhancer labels a broad diversity of GABA-INs and differences in functional properties for inhibitory cell types have been observed in mature ferrets^{8,17}. Because of **biological** limitations in identifying cells based on molecular markers early in development^{45,46}, we pooled all GABA-INs. Importantly, we did not observe significant heterogeneity in cellular responses of Naive animals, suggesting that most GABA-INs do not exhibit modular responses or orientation tuning early in development **and that any differences in the developmental progression of GABA-IN subclasses labeled by the mDlx enhancer likely are not reflected in the network response to simple stimuli. Indeed, we are optimistic that novel methods targeting interneuron subtypes will allow for the assessment of subtype-specific developmental trajectories of GABA-INs in the future⁴⁷. However, our findings, to our knowledge, are the first observations of changes in the evoked patterns of activity of GABA-INs in a modularly organized cortex and serve as a basis for the design of future studies investigating subtype-specific changes in GABA-INs across development.***

Reviewer #2 (Remarks to the Author):

The authors examined developmental changes of visual responses of interneurons (INs) after eye-opening in the ferret V1 and found that orientation selectivity is poor at eye-opening and matured several days after eye-opening depending on the visual experience. At the population level,

widespread activity was observed at eye-opening. The widespread activity reduced independent of the visual experience and clustered activity emerged after several days of visual experience, suggesting the visual experience-dependent and independent mechanisms of the development. Together with the previous study about excitatory cells, the authors suggest the delayed maturation of the inhibitory network.

Although the argument of delayed maturation of inhibitory neurons has already been reported in the mouse V1 (Kulman, Trachetenberg et al., 2011), investigation in the animal with functional columns is a very important theme and attractive for many readers in the sensory neuroscience field. The authors further reveal the developmental change of population activity pattern in INs which has not been previously reported. However, I feel that there are several concerns that should be addressed before the publication. Additional data and analyses would improve the manuscript.

We thank the reviewer for their positive comments regarding our manuscript. Below we outline point-by-point the changes we have made that address the concerns raised.

Major points

1. The authors reported widespread, visually evoked activity in naïve animals. I feel that the descriptions about the widespread activity is not enough to understand what kinds of activity was actually observed in naïve animals. Please more clearly describe activity patterns in naïve animals. First, I wonder whether this activity is visually evoked or reflects spontaneous activity which occurs during visual stimuli. For example, analyzing the reliability of onset timing should be useful to reveal this point.

Please refer to our responses to Reviewer #1 Comments 1 and 2.

Second, please demonstrate more examples of activity patterns (e.g., raster plot shown in Fig2a, activity pattern map in supplementary figure 2b, a movie of the widespread activity, etc.).

We have added trial responses for Naive, Brief, and Extended animals to our new *Supplementary Fig. 3e*, and trial responses for Binocular Deprivation and Recovery animals in *Supplementary Fig. 4f*. These additional plots further underscore the broad participation in visually evoked responses seen in Naive animals, compared to the clustered responses observed in Brief and Extended animals. These raster plots further demonstrate how visually evoked responses are not merely driven by spontaneous activity, as visually evoked responses are locked to stimulus presentation and are larger than the responses expected from spontaneous activity as demonstrated by the blank trials (the last 10 trials of the raster plots).

In addition, we have added single trial widefield responses for Naive, Brief and Extended experience animals in our new *Supplementary Fig. 3a*, and Binocular Deprivation and Recovery animals in *Supplementary Fig. 4a*. We hope that our added figures demonstrating single trial responses further demonstrate the lack of observed clustered and modular activity in Naive animals as compared to Brief and Extended animals and observed clustering of activity in BD and Recovery animals. In sum, these changes further demonstrate the lesser contribution of clustered and modular responses to GABA-IN network responses in Naive animals.

Third, it is not clear how often the widespread visually evoked activity is observed. For example, in the activity pattern of naïve animal shown in the supplementary figure 2b, population response patterns highly varied across trials and the pattern looks clustered in some trials: widespread activity across all cells is observed in the trial 1 and 3, some clustered activity in trial 4, and almost no activity in trial 2 and 5. According to these results, pattern correlation would not be so high in naïve animals. However, the pattern correlation matrix shows very high values across trials/stimuli (figure 2b), suggesting the pattern of widespread activity is consistent across trials. I am puzzled by this inconsistency. When the authors calculate the pattern correlation (figure 3b), the average activation across cells is subtracted (by definition of the correlation), which might underestimate the amplitude variability. Analyses of population-averaged activity and participation rate may be informative.

We thank the reviewer for their comments. Indeed, the trial-to-trial pattern correlations as computed for Fig. 2b are likely to eliminate the contributions of widespread activity. What these elevated correlations reflect, however is the residual heterogeneity of responses that are reflected by the weak clustering of locally correlated activity as demonstrated in Fig 3g. Indeed, we do not use the trial-to-trial pattern correlation as a measure of the widespread, correlated activity precisely for this reason. Instead, our analysis uses the trial-to-trial pattern correlation to capture the fine-scale differences in cellular activity and relate those differences to oriented stimuli. The data that supports our observation of widespread, correlated activity is the elevated cellular pairwise correlation, for both short- and long-range pairs as shown in Fig 3g.

How could widespread activity result in correlated activity when the average cellular activity is subtracted during the correlation? Unreliable activation of the inhibitory network. This is evident in the single trial examples we show for Naive animals, where the degree to which cells are active is inconsistent. To further highlight the unreliability of inhibitory responses as well as the widespread activation of GABA-INs to visual stimuli, we have added example raster plots of cellular activity for Naive, Brief and Extended experience animals (Supplemental Fig. 3e). These raster plots demonstrate how the activity of GABA-INs changes across development. In Naive animals, responses are highly variable, and frequently large fractions of the inhibitory population are active together. In contrast, Brief and Extended experience animals have orientation tuned responses for small groups of cells and lack trials where large fractions of the inhibitory population are co-active.

We acknowledge that previously we referred to this as “widespread” or “uniform activity”, and that may have contributed to the confusion about how inhibitory networks respond to visual stimuli early in development. Instead, the modulation of the cellular activity with visual stimuli is more strongly coupled for GABA-INs across short and long ranges in Naive animals. References to the nature of early inhibitory network responses have been changed to “widespread, correlated activity” in our manuscript. We hope that this new phrasing better captures the nature of the GABA-IN responses we observed early in development and clarifies how the pairwise-response correlation maps can have elevated positive correlations.

Fourth, related to the above point, I am interested in how often the clustered patterns are observed in naïve animals. If the clustered activity is observed with a substantial frequency in naïve animals, it may be a seed of the clustered pattern observed in a later stage of development. Is the clustered

activity pattern variable across trials? Another possibility is that clustered activity is already established in naïve animals, but it was masked by a widespread activity. If the widespread pattern is removed and pattern correlation is computed for the remaining data, is the resulting pattern correlation matrix similar to that in observed in the matured animals?

As the reviewer has previously observed, the trial-to-trial pattern correlation matrix by the nature of the correlation, removes the uniform component of the pattern responses. The trial-to-trial pattern correlation matrices, however, are not consistent with those observed for animals with visual experience. First, even though there is a degree of fine-scale structure in the response, the response is not reliably activated by oriented stimuli. In addition, we fail to see anti-correlated trials for even a fraction of the orthogonal trial-pairs in Naive animals (Fig. 2b), suggesting that even filtering out large events would not reveal a trial-to-trial pattern correlation matrix like that observed for Brief experience. Thus, while widespread activity may mask some of the effects of clustered activity in GABA-INS early, it is likely not the sole factor contributing to the unreliable activation of the inhibitory network early in development.

Indeed, patterns of activity in Naive animals, do show a degree of clustering as evidenced by the difference in overall correlations for both neighboring and distant neurons (Fig. 3g), and as the reviewer correctly points out this clustering can be observed on some trials (Supplementary Fig. 3d,e). The difference in the correlation for short- and long-range cell pairs, however, is small in comparison to that observed for Brief experience and show an elevation in overall correlation (Fig. 3g). Coupled with recent observations of modular patterns of spontaneous activity in GABA-IN networks⁵, this does suggest that, visually evoked widespread, correlated activity could mask clustered activity. This clustered activity may reflect underlying structure in GABA-IN networks. Despite this, modular responses are reliably evoked by stimulus presentation in Brief and Extended animals. These findings highlight that widespread, correlated activity may mask underlying network structure and underscore how the relative responses to visual stimuli with GABA-IN network changes across development.

We have updated text within our manuscript to make more explicit that modular responses in GABA-INS may be masked by widespread, correlated activity early in development (lines 262-267):

*Our findings suggest that correlated responses in naive inhibitory networks are only modestly greater for neighboring cells than those at greater distances, and **widespread, correlated activity** across the network dominates responses. **This correlated activity may mask modular network structures of GABA-IN networks.** After eye opening, the correlation of neighboring cells remains high and long-range correlations decrease, resulting in **observable** patchy, clustered patterns of activity (Supplementary Fig. 3c-e).*

2. I am curious about the spontaneous activity of INs at eye-opening, its developmental change, and its relationship to the visually evoked activity. The authors' group has previously reported that spontaneous activity of mainly excitatory neurons (GCaMP expressing cells under synapsin promoter) is spatially clustered and related to the functional column at eye-opening (Smith et al., 2018). Thus, it should be important to examine whether the spontaneous activity of INs at eye-opening is also widespread as the visually evoked activity or spatially clustered as the excitatory cells. Revealing the similarity/dissimilarity between spontaneous and sensory-evoked activity should provide an

important clue to speculate underlying mechanisms. In addition, developmental change of IN's spontaneous activity and its comparison to that of excitatory cells is an important theme and it must increase the impact of this manuscript.

Please refer to our response to Reviewer #1 Comment 3 for details about spontaneous activity of GABA-INs across development. While we agree that developmental changes in spontaneous activity relative to evoked activity and the relationship between excitatory and inhibitory neurons is an important theme that merits further investigation, we chose to focus on the visually evoked responses of GABA-INs across development and the role visual experience plays in shaping these responses. In doing so we observed multiple changes including the development of orientation tuning, reduction in response variability and the reduction in widespread, correlated activity across the GABA-IN population and distinguished the role of experience in these observed changes and highlight the differences in developmental trajectories of GABA-INs and excitatory neurons. Together we believe these observations comprise a significant step forward in our understanding of the development of GABA-INs.

3. I think that the difference in correlations for short- and long-range cell pairs (figure 3g and 5f) is not a good measure of modularity. It just shows the existence of some clustering but does not show the modular periodical pattern of functional maps. Please consider other measures of modularity.

While local clustering captures a necessary component of modular patterns of activity, we acknowledge that local clustering alone is not sufficient for modular organization. Thus, we have updated our manuscript with a new analysis, based on a previously established method fitting of Morlet wavelengths in order to quantify the characteristic column spacing (wavelength) of pixelwise correlation maps. We have included this additional analysis in Supplementary Fig. 2b and provided data for reference wavelengths at each age for excitatory neurons as labeled by the hSyn promoter (which as previously been demonstrated to primarily label excitatory neurons⁷). We have updated our text to incorporate this wavelet analysis, and the updated text is reproduced for reference below (lines 224-234):

To further quantify the modularity of the correlation maps, we modified a method for measuring the column spacing (wavelength) of modular networks¹⁹. We observed comparable wavelengths for excitatory neurons and GABA-INs in Brief and Extended experience animals (Supplementary Fig. 3b; Brief excitatory 0.736 ± 0.032 mm vs GABA-INs 0.770 ± 0.053 mm, $n=6$ vs 4 , $p=0.5120$; Extended excitatory $0.797 \pm .057$ mm vs GABA-INs 0.844 ± 0.041 mm, $n=5$ vs 5 , $p=0.4730$; Mean \pm SEM, bootstrap test). In contrast, GABA-INs in Naive animals had a significantly larger wavelength (Naive excitatory 0.779 ± 0.020 vs GABA-INs 0.924 ± 0.026 , $n=7$ vs 5 , $p=0.0048$, Mean \pm SEM, bootstrap test), suggesting that responses are not as well associated with excitatory modular networks at eye opening. Thus, naive inhibitory networks do not have an apparent modular organization at the millimeter-scale for evoked activity, and after a week of visual experience modular organization of responses and orientation preferences emerge.

4. Based on the results of binocular deprivation (BD), the authors argue that the development of orientation tuning depends on the visual experience, whereas the reduction in the widespread

responses is independent of the visual experience. “the difference in correlations for short- and long-range cell pairs was similar when compared to animals with normal extended experience (BD 0.12 ± 0.03 vs Extended 0.19 ± 0.02 , $p=0.0630$, $n= 4$ vs 4 , Mean \pm SEM, bootstrap test).” However, the difference in correlations decreases from 0.19 to 0.12 (see also fig 5f and fig 3g), although the difference is not marginally significant ($p=0.06$). I wonder whether the reduction in the widespread activity in BD animals is really the same as the reduction in the naturally reared animals and whether the reduction in the widespread responses is completely independent of the visual experience. Further, as pointed out in 3), I do not think that the difference in correlations for short- and long-range cell pairs is not a good measure of modularity. Only with this index, I do not think the modular pattern in the BD animal is firmly supported. Please demonstrate the correlation maps in BD animals in figure 5.

We regret that our manuscript implied that we observed modular responses based on the cellular observations, and to reduce confusion we have removed the paragraph in our manuscript that addressed the link between weak orientation tuning and variable modular responses. In its place we have added new analyses that more directly address the modular organization of GABA-IN at the millimeter scale and expanded our analysis of cellular-scale clustering of responses to provide more nuanced insights into the changes that occur with after binocular deprivation.

Our first analysis provides more information about the organization of GABA-IN responses at the millimeter scale. We have included new epifluorescence analysis with example single trial responses (Supplementary Fig. 3a), and pairwise pixel correlation maps (Supplementary Fig. 3b) for Binocular Deprivation and Recovery animals, which we hope more clearly demonstrates the qualitative difference in responses for Binocular Deprivation animals. We have furthermore quantified the wavelength of modular activity in GABA-INs for Binocular Deprivation and Recovery animals labeled for excitatory or inhibitory neurons (Supplementary Fig. 3c). No difference was observed in the wavelength of pairwise pixel correlation maps for excitatory neurons and GABA-INs in either the Binocular Deprivation or Recovery animal groups. These findings suggest that delayed onset of experience does not merely pause the development of GABA-INs, but instead the widespread, correlated activity observed early in development is reduced unmasking modular network organization. We have updated our text to include this analysis and duplicated these changes below for your convenience.

Updated text (lines 342-354):

These results indicate that experience plays a key role in the development of orientation selective responses in GABA-INs, and one might expect that experience plays a similar role in the development of modular spatial organization for GABA-INs. First, we assessed whether large-scale organization of patterns was observable in BD and Recovery animals, using widefield epifluorescence imaging. Patterns of activity in both BD and Recovery animals displayed clustered, modular organization (Supplementary Fig. 4a,b). Consistent with our widefield observations, we observed diversity in the patterns for cellular trial activity (Supplementary Fig. 4d-f) in BD animals compared to Recovery animals. We further assessed the modularity of the pairwise pixel response correlations by computing the column spacing. Both BD and Recovery animals had comparable wavelengths to excitatory neurons (Supplementary Fig. 4c; BD excitatory 0.867 ± 0.065 mm vs GABA-INs 0.897 ± 0.064 mm, $p=0.6974$, $n=4$ vs 4 ; Recovery excitatory 0.796 ± 0.046

vs GABA-INs 0.798 ± 0.031 mm, $p=0.9618$, $n=5$ vs 4 ; Mean \pm SEM, bootstrap). Thus, large-scale modular organization in GABA-INs appears independent of visual experience over the first week following normal eye opening.

Next, we have added new analyses and expanded our discussion regarding the observed changes at the cellular scale after binocular deprivation. Specifically, have added new comparison for the strength of responses correlations observed for short- and long-range cellular pairs for BD animals as compared to age matched animals with Brief experience. While our observation of an overall reduction in response correlations hold when BD animals are compared to Naive animals, Brief experience animals still maintain a high level of responses correlation for short-range cellular pairs as compared to BD animals, suggesting that delay of the onset of visual experience could impact the strength of local correlations.

Updated text (lines 355-376):

How well is the large-scale organization of GABA-INs reflected in the cellular response? We assessed the spatial organization of cellular response correlations (Fig. 5e) and found that both BD and Recovery animals demonstrated significantly higher correlations for short-range (<100 μ m) compared to long-range (300-400 μ m) cell pairs (Fig. 5f, BD short 0.25 ± 0.08 vs long 0.13 ± 0.05 , $p=0.0357$, $n=4$; Recovery short 0.26 ± 0.02 vs long 0.09 ± 0.02 , $p=0.0028$, $n=4$; Mean \pm SEM, Student's paired t-test). The difference in correlations for short- and long-range cell pairs were like animals with normal extended experience (BD 0.12 ± 0.03 vs Extended 0.19 ± 0.02 , $p=0.0630$, $n=4$ vs 4 , Mean \pm SEM, bootstrap test). Importantly, correlations for both short- and long-range cell pairs were systematically lower for BD animals when compared to Naive animals (Naive short 0.49 ± 0.07 vs BD short 0.25 ± 0.08 , $p=0.0392$; Naive long 0.36 ± 0.08 vs BD long 0.13 ± 0.05 , $p=0.0448$; $n=4$ vs 7 , Mean \pm SEM, bootstrap test). However, BD animals had a significantly lower correlation for neighboring GABA-INs compared to age-matched animals with Brief experience (BD short 0.25 ± 0.08 vs Brief short 0.43 ± 0.03 , $p=0.0392$; BD long 0.13 ± 0.05 vs Brief long 0.04 ± 0.03 , $p=0.1012$; $n=4$ vs 6 , Mean \pm SEM, bootstrap test). Thus, the widespread, correlated activity that dominates responses in Naive animals was greatly reduced in BD animals, but short-range correlations were impaired. In Recovery animals, subsequent experience did not affect the difference in short- and long-range correlations (BD 0.12 ± 0.03 vs Recovery 0.17 ± 0.02 , $p=0.1572$; $n=4$ vs 4 , Mean \pm SEM, bootstrap test), suggesting delayed visual experience does not alter the degree to which responses to oriented stimuli cluster. Together, our findings demonstrate that the reduction of widespread, correlated activity and subsequent unmasking of clustered activity in inhibitory networks can develop in an experience-independent manner over the developmental period spanning the first week of normal visual experience. However, the maintenance of short-range correlations early in development and orientation associated modular responses requires visual experience.

We thank the reviewer for their feedback, and believe these new analyses further highlight the nuanced changes that occur after binocular deprivation and strengthen our conclusions about the role visual experience plays in the development of orientation tuned responses for GABA-INs.

5. The authors examined the difference in the monocular orientation preference mismatch between Brief and Extended, but not Naïve. Although the percentage of the orientation-selective cells in naïve animals is smaller (around 25%), it should be possible to examine the monocular orientation preference mismatch in these cells. The authors suggest that “inhibitory and excitatory neurons have fundamentally different developmental progressions for orientation selectivity and binocularly aligned responses”, but it is strange to compare naïve and brief for excitatory neurons and brief and extended for inhibitory neurons. In order to suggest that the binocularly aligned responses are also different in inhibitory neurons, it is necessary to analyze the binocular mismatch in inhibitory neurons in naïve animals

We have previously demonstrated that the binocular alignment of orientation preference in the ferret visual cortex is a network-scale reorganization of patterns of activity. Thus, we choose to run our analysis on a per animal basis, where the mean monocular mismatch of cellular orientation preferences represents a representative measure of the misalignment of the entire network. As the reviewer points out, ~24% of GABA-INs display significant orientation tuning for binocular responses. A smaller fraction of neurons, however, display orientation tuning for contralateral (~14%) or ipsilateral stimulation (~20%) and these populations were not always overlapping. Given the smaller number of cells labeled by the mDlx enhancer and the small fraction of neurons that are well-tuned for both binocular and monocular visual stimuli, we were unable to adequately assess the misalignment of monocular and binocular network preferences in Naive animals.

Therefore, we chose to compare only the Brief and Extended experience age groups for binocular alignment for GABA-INs. While these age groups are not exactly the same as those compared in our previous study, we believe that these age groups compare relative periods of development of orientation tuning for excitatory neurons and GABA-INs. Our analyses highlight that at the age where the majority of GABA-INs (Brief) have orientation preferences, preferences are well matched for both eyes and the observed difference does not significantly decrease with subsequent experience (Extended). In excitatory neurons, at the age where the majority of excitatory neurons display orientation preferences (Naive), significant differences in orientation preferences are observed for the eyes, and subsequent experience (Brief) does align orientation preferences. These differences highlight how the developmental trajectories of GABA-INs and excitatory neurons is not merely delayed, but also fundamentally different.

6. The clustered activity in the extended group was relatively weak compared to the brief group (e.g., Figure 3b-g), and the authors speculate this could result from the diversification of neuronal preference and/or cortical expansion (L216-217). Another possibility is an artifact resulting from the reduction of visual responses due to excessive expression of GCaMP. Please exclude this possibility.

Overexpression of GCaMP due to prolonged viral expression is a confound that could alter the responses of neurons. We do not believe, however, that this effect contributes to the lower cellular-pairwise correlations we observed in Extended experience animals. First, we did not observe a significant increase in the number of neurons with nuclear filling of GCaMP6s, a marker of cytotoxicity

due to overexpression. Were overexpression a possible mechanism driving lower visual responsiveness, we would expect to see a larger fraction of neurons that failed to achieve our threshold for visual responsiveness. Indeed, we failed to observe a significant change in the fractions of neurons that were non-responsive in Extended experience animals (Fig 1i). Despite the low number of non-responsive neurons, we excluded any non-responsive cells from our correlation analyses for all age groups, in order to rule out the possibility that low correlations are due to lack of responsiveness.

One possible explanation that we highlighted in our manuscript, is the development of neuronal preferences for other relevant dimensions such as spatial or temporal frequency. This increased specificity could result in cellular response heterogeneity, a neuronal network feature that has previously been theorized to play a role in the efficient coding of stimuli⁸. We have added a new citation to further highlight how the reduction in local correlations may result from normal developmental processes that are optimizing the coding of visual stimuli.

We have updated the text to address these points (lines 253-267):

*We also observed a significant reduction in the clustering between animals with Brief and Extended experience (Fig. 3g, Brief 0.40 ± 0.06 vs Extended 0.19 ± 0.02 , $p=0.0106$, bootstrap test). **Loss of responsivity due to the overexpression of GCaMP6s in Extended experience animals is likely not the cause of the observed reduction in clustering as a large fraction of GABA-INs remained visually responsive for Brief and Extended experience animals (Fig. 1i). Lower response correlations could result from cortical expansion and/or from diversification of neuronal preferences along other relevant dimensions (such as spatial or temporal frequency). Diversification of neuronal preferences results in increased cellular responses heterogeneity, which has been theorized to contribute to the efficient coding of stimuli²⁰, and thus could reflect further optimization of visual representations in the second week of visual experience. Our findings suggest that correlated responses in naive inhibitory networks are only modestly greater for neighboring cells than those at greater distances, and widespread, correlated activity across the network dominates responses. This correlated activity may mask modular network structures of GABA-IN networks. After eye opening, the correlation of neighboring cells remains high and long-range correlations decrease, resulting in observable patchy, clustered patterns of activity (Supplementary Fig. 3c-e).***

Minor point

The authors only demonstrate the data of INs in the manuscript. As commented above, the authors' group has reported the data of (mainly) excitatory cells. Direct comparison between excitatory and inhibitory cell is informative and improve the impact of this manuscript. The schematic diagram which directly compares the result of INs to those of excitatory cells in literature is very useful for readers who are not familiar with the topics to recognize the results of this manuscript.

We agree that a schematic of the developmental timeline for both GABA-INs and excitatory cells may prove to be useful for readers who are not familiar with the topic. Thus, we have added Fig. 6 to our manuscript outlining the developmental progression of excitatory and inhibitory neurons in the visual cortex around the time of eye opening and provided an expanded discussion of the differences in

developmental trajectories of excitatory and inhibitory neurons in our discussion. We hope that this figure and additional discussion will make clear the delayed nature of the development of orientation selectivity and modular evoked patterns of activity in interneurons as compared to excitatory neurons.

Updated text (lines 391-400):

Excitatory and inhibitory neurons in the visual cortex in the period around eye opening undergo distinct developmental trajectories (Fig. 6). At eye opening excitatory networks display modular patterns of activity for both spontaneous and evoked activity, and initial cellular orientation preferences are established^{4,6,7}. Recent studies have suggested that like excitatory neurons, GABA-INs also display modular patterns of spontaneous activity at eye opening²¹. In contrast, we show that modular patterns are not evident for stimulus evoked activity and orientation tuning is not well established for GABA-INs at eye opening. Over the first week of normal visual experience, excitatory neurons continue to develop orientation selectivity, binocular orientation preference alignment and direction selectivity¹⁻⁴. Our data demonstrate that during this period experience-dependent processes drive the development of orientation selectivity, binocular alignment, and direction selectivity.

In figure 3b and c, the map area in a naïve animal is significantly smaller than other groups. Why is the area in naïve animals small? In figure 3e, pixel-pixel distance is plotted up to 1.5 mm. Thus, there should be some data in naïve animals which cover more than 1.5 mm. The small size of the correlation map in figure 3c prevents the readers to evaluate the modular structures. I suggest demonstrating the wider maps in naïve animals.

Differences in the size of the ROIs in Fig. 3b,c reflect normal variation in the extent of GCaMP6s expression. While the ROI shown for the naïve animals is smaller, it still extends beyond the 1.5 mm in multiple directions. Furthermore, our new wavelength analysis (Supplementary Fig. 2b) suggests that if modular network structure is evident in the Naïve GABA-IN networks it should be on the scale of 0.7-0.8 mm and should be evident within the ROI. Therefore, the small size of the ROI shown in Fig. 3 does not fundamentally alters any of our analyses.

The monocular orientation mismatches in inhibitory neurons in Brief animals (8.54 degree) is smaller than those in excitatory neurons (12.73 degrees). Does the author have any suggestion about why the mismatch is smaller in inhibitory neurons?

We acknowledge that our observed monocular mismatch in inhibitory neurons at first glance appears to be lower than that of our previously observed mismatch for excitatory neurons. One possible factor that could have impacted our observed monocular mismatch, is a difference in the methods we used to assess orientation preferences. Previously, for excitatory neurons, we fit orientation tuning curves for responses in orientation space (i.e. we ignored the direction of the drifting grating), because at the age we were assessing orientation preferences, direction selectivity is underdeveloped for excitatory neurons. In the present study, in order to assess orientation preferences, we fit direction tuning curves to neuronal responses. We chose to do this because in Extended experience animals, we expected a proportion of GABA-INs to display significant direction selectivity. An additional factor that

could also impact the accuracy of our fitted tuning curves is the broadness of the tuning, and we would expect that GABA-INs would have broader tuning curves based on the lower OSI observed by *Wilson et al.*⁷.

Indeed, normal errors associated with response variability can impact the assessed orientation preferences for neuron can give rise to a large monocular mismatch. In *Chang et al.*, we assessed this error by resampling responses to monocular orientation presentation. This gave us a bound on the observed difference that can be attributed to response variability. Using this method, we found that the median observed difference in orientation preference was 12.21 and 12.34 degrees for the contralateral and ipsilateral eyes respectively⁴.

To properly discern whether our observed difference result from differences in fitting accuracy or random sampling, monocular mismatch would need to be calculated for excitatory and inhibitory neurons that are labeled within a shared field of view. Accounting for this difference does not alter our findings, as we did not make any statistical tests comparing GABA-INs and excitatory neurons. Instead, we compared inhibitory neurons for animals with Brief and Extended experience using a common analysis, and we failed to observe significant differences between Brief and Extended experience animals. This suggests that in Brief experience animals, when the majority of cells are orientation selective, GABA-INs have matched orientation preferences for binocular and monocular stimulation, and that these orientation preferences remain well matched.

I could not find how the authors calculated the pattern correlation matrix. The description in the method is very sketchy. The trial number of the pattern correlation matrix seems to correspond to the direction of stimuli because adjacent trials tend to show high correlations. Please clearly describe this point in the legend.

We apologize that our method for generating pattern correlation matrices was unclear. As the reviewer has pointed out, trials were sorted by the stimulus direction, thus trials along the diagonal correspond to matched directions. We have added a note about this in the appropriate figure legends and updated the methods. Excerpts from these changes are shown below:

Fig. 2:

Fig. 2 Population responses to orientation become more differentiable after visual experience. (a) Trial-to-trial pattern correlation matrices were generated by comparing population trial responses across stimulus conditions. *Trials were sorted by direction, such that each group of 10 trials corresponds to the repeated presentation of a stimulus direction. Sixteen directions were presented equally spaced between 0 and 337.5 degrees.* (b) Example trial-to-trial pattern correlation matrices for a Naive (left), Brief (middle), and Extended (right) experience animal. *Dashed lines denote groups of the same stimulus direction.* (c) Comparisons of trial-to-trial pattern correlation for matched (red) and orthogonal (blue) stimuli across age groups (Naive: N (n=7), Brief: B (n=6), Extended: E (n=5)). (d) Schematic of the template decoding algorithm. (e) Decoding fraction versus number of cells used in the template decoder for Naive (left, n=7), Brief (middle, n=6), and Extended (right, n=5) animals. Results for individual animals (grey), grouped average (black), and shuffle (red). (f) Change in decoding accuracy versus the cell number in the

decoder for Naive (blue, n=7), Brief (magenta, n=6), and Extended experience (gold, n=5). Shaded regions denote SEM. *: p<0.05, **: p<0.005, ***: p<0.0005

Fig. 5:

Fig. 5 Reduction in spatially uniform responses occurs without visual experience. (a) Trial-to-trial pattern correlations for a Binocular Deprivation (left) and Recovery (right) animal. *Trials were sorted by direction, such that each group of 10 trials corresponds to the repeated presentation of a stimulus direction. Sixteen directions were presented equally spaced between 0 and 337.5 degrees.* Dashed lines denote groups. (b) Comparison of matched (red) and orthogonal (blue) trial-to-trial pattern correlations for Binocular Deprivation (n=4) and Recovery (n=4) animals. Individual animals are shown as gray and error bars denote SEM. (c) Template decoding accuracy for Binocular Deprivation (n=4) and Recovery (n=4) animals for different template sizes. Red line indicates chance decoding levels, black shows average decoding level, and gray lines denote individual animals. (d) Change in decoder accuracy for the nth cell included in the template decoder for Binocular Deprivation (black, n=4) and Recovery (green, n=4) animals. Shaded regions denote SEM. (e) Example cellular pairwise correlations for Binocular Deprivation (left) and Recovery (right) animals. The seed cell is denoted by the green triangle and scale bar denotes 100 μm . (f) Binocular Deprivation (n=4) and Recovery (n=4) differences of cellular pairwise response correlations for short- (black, < 100 μm) and long-range (green, 300-400 μm) cells pairs. Mean \pm SEM. Gray points denote measurements for individual animals. ***: p<0.0005, ns: not significant

Methods (lines 576-581):

Trial-to-trial pattern correlations were computed by first sorting trials by the direction of the drifting grating, then taking all cells within a field-of-view and computing Pearson's correlation coefficients across trial pairs. Matched and orthogonal trial-to-trial correlations were computed as the mean correlation for matched and orthogonal stimuli. We chose to perform these comparisons in direction space to minimize differences in evoked patterns due to increasing direction selectivity across development.

I could not find the description of the spatial frequency in the section of Visual stimulation (L464). Please describe it. Is there any possibility that poor visual response in naive animals is due to non-preferred spatial and/or temporal frequency of grating stimuli? Please exclude this possibility.

We regret the omission of this important detail from our methods section. We have updated the text in the methods to address this. While changes in preferences for spatial and temporal frequencies likely do occur over this developmental period, we purposely chose slow moving and low spatial frequency visual stimuli to optimize responses in young animals. The updated methods text has been duplicated below (lines 521-525):

To evoke orientation-specific responses, 0.1 cycles per degree full-field square gratings at 100% contrast drifting at 1 Hz were presented in 16 directions. Low spatial and temporal frequencies were chosen to optimize responses in young animals.

In figure 3c, the seed point should have a correlation value of 1. However, in the extended animals, the value seems smaller than 1 (although the seed point itself is hidden). Why?

Seed points for the correlation maps shown in Fig. 3c have a correlation value of 1, by definition, as the reviewer has pointed out. In our maps however, we designate the seed point as a green square in order to distinguish the seed point from other regions in the map that have high levels of correlated activity with the seed point. The apparent lower correlations observed in the widefield correlations maps for extended experience, are a likely a reflection of the overall lower cellular pairwise correlations for shorter range cell pairs that are also evident in the two-photon microscopy and shown in Fig. 3g. We have reduced the size of the marker designating the seed points in Fig. 3c, and in the new Supplementary Fig. 3b. We hope that these changes will make the local correlations around the seed point clearer.

In figure 4, data of the brief group should be plotted, and a comparison between BD and the brief group should be demonstrated in the figure, although they presented the results in the manuscript.

To aid with the comparison between BD brief experience, we have added in a reference line to denote the levels associated with Brief Experience to Fig. 4e-g.

In the template matching analysis, please clarify that decoder performance was tested with data from the trials that were not used to make the templates.

We thank the reviewer for pointing out the lack of clarity regarding the testing of our template decoder. We have amended the text in the Methods to clarify that the decoding accuracy was based on trials that were not included in the generation of the templates. Changes to the text are included below (lines 582-590):

*Finally, we implemented a normalized template matching algorithm as previously described⁴. Briefly, template population responses for each stimulus condition were generated by computing the median response using half the trials across all direction conditions. We assessed the performance of the template decoder **by predicting the presented stimuli for the trials not used to generate the template population responses** and altered the number of cells included in the templates to generate template decoding accuracy versus template size. To assess the overall performance of the template decoders, we chose to use 40 cell templates, as the accuracy across all conditions appeared to approach peak accuracy near template size. To assess chance decoding rates, we generated templates using shuffled trials and assessed the decoding accuracy of a shuffled test data set (n=1000).*

“Age matched animals with normal visual experience also displayed higher similarity of response patterns for matched stimuli, and lower similarity for orthogonal stimuli (matched Recovery 0.81 ± 0.02 vs Extended 0.76 ± 0.02 , $p= 273 0.1118$; orthogonal Recovery 0.25 ± 0.02 vs Extended 0.09 ± 0.02 , $p=0.0076$; $n= 5$ vs 4 , Mean \pm SEM, 274 bootstrap test).”

“lower similarity” should be “higher similarity”.

We apologize for the error in phrasing, we have amended the text to make clear we are highlighting the high degree of similarity for population responses to matched stimuli for both the Recovery and Extended cohorts, which are not significantly different. Updated text is shown below (lines 317-320):

*Age matched animals with normal visual experience also displayed **comparable, high** similarity of response patterns for matched stimuli, and lower similarity for orthogonal stimuli (matched Recovery 0.81 ± 0.02 vs Extended 0.76 ± 0.02 , $p= 0.1118$; orthogonal Recovery 0.25 ± 0.02 vs Extended 0.09 ± 0.02 , $p=0.0076$; $n= 5$ vs 4 , Mean \pm SEM, bootstrap test).*

L107. Reference number 15 seems to be wrong.

L672. Reference number should be 45.

L583. Chang, J. T., Whitney, D. & Fitzpatrick, D. Experience-Dependent Reorganization Drives Development of a Binocularly Unified Cortical Representation of Orientation. Neuron 0, (2020). Please add the volume number and page number.

We have updated our references to correct for the errors and omissions highlighted above.

References

1. Chen, T.-W. *et al.* Ultra-sensitive fluorescent proteins for imaging neuronal activity. *Nature* **499**, 295–300 (2013).
2. Li, Y., Fitzpatrick, D. & White, L. E. The development of direction selectivity in ferret visual cortex requires early visual experience. *Nat. Neurosci.* **9**, 676–681 (2006).
3. White, L. E., Coppola, D. M. & Fitzpatrick, D. The contribution of sensory experience to the maturation of orientation selectivity in ferret visual cortex. *Nature* **411**, 1049–1052 (2001).
5. Mulholland, H. N., Hein, B., Kaschube, M. & Smith, G. B. Tightly coupled inhibitory and excitatory functional networks in the developing primary visual cortex. *eLife* **10**, e72456 (2021).
6. Gao, W. J., Wormington, A. B., Newman, D. E. & Pallas, S. L. Development of inhibitory circuitry in visual and auditory cortex of postnatal ferrets: immunocytochemical localization of calbindin- and parvalbumin-containing neurons. *J. Comp. Neurol.* **422**, 140–157 (2000).
7. Wilson, D. E. *et al.* GABAergic Neurons in Ferret Visual Cortex Participate in Functionally Specific Networks. *Neuron* **93**, 1058-1065.e4 (2017).
8. Chelaru, M. I. & Dragoi, V. Efficient coding in heterogeneous neuronal populations. *Proc. Natl. Acad. Sci.* **105**, 16344–16349 (2008).

REVIEWER COMMENTS

Reviewer #1 (Remarks to the Author):

The authors have satisfactorily addressed my previous concerns.

Reviewer #2 (Remarks to the Author):

In the new manuscript, the authors added additional data, most of which support the author's arguments and significantly improve the manuscript. However, I still have a few concerns. I recommend that these points should be addressed before publication.

1) In the new manuscript, the author performed a wavelet analysis to evaluate the modular activity patterns in the correlation map of the wide-field calcium imaging and reported the wavelength as a modularity metric. However, the wavelength indicates a distance between modules if the modularity exists, but does not support the existence of the modularity. Thus, the wavelength is not a good metric to confirm the modular patterns. Although the wavelength tells us the size of the modular organization if the wavelet fitting is good, some index for the existence of the modularity should be considered. When the modularity does not exist as the authors claim for Naïve, quantification of the wavelength would be meaningless,

2) Related to 1), in the Supplementary fig 4c, the authors reported the comparable values of the wavelength between excitatory and inhibitory cells in the BD and argue that large-scale modular organization in GABA-INs appears independent of visual experience (L353-354). However, in order to claim that the modular organization in GABA-INs is independent of visual experience, the wavelength in GABA-INs (and excitatory cells) should be compared with that in the normally reared, time-matching animals (i.e., Brief). Then, the wavelength of GABA-INs in BD appears higher than that in Brief and similar to that in Naïve (Supplementary fig. 3). This seems to suggest that the wavelength of modular pattern is sensory-dependent, which is inconsistent with the author's argument. The representative correlation map for BD shown in the supplementary fig 4b also seems less clear than the clear modular structure for Brief shown in fig 3c.

3) Insufficient CCaMP expression resulted in the relatively small size of ROI (i.e., small size of robust visual evoked area, L592-593) in the wide-field imaging in the naïve, which does not completely discard the possibility that the modular pattern of visually evoked activity is missed in Naïve. The modular pattern of spontaneous activity in Naïve has recently been reported (Mulholland et al 2021). If the authors have observed the modular pattern of spontaneous activity during blank or baseline period in Naïve even within this small size of ROI, it would be really helpful to claim that the modular pattern

cannot be missed even within this small size of ROI. This is optional, but descriptions about the spontaneous activity should ensure data quality and consistency with the published data, and also demonstrate a clear contrast between spontaneous and visually induced activity patterns which makes the author's argument more convincing. Alternatively, as commented in the author's reply, it would be helpful if the authors address in the manuscript that the small size of the ROI does not fundamentally alters any of their analyses.

4) In Naïve animals, the visually evoked activity in GABA-INS shows wide-spread patterns, although the spontaneous activity is modular (Mulholland et al 2021). It would be helpful if the authors discuss what kind of circuit mechanisms can cause this difference.

5) The authors claim "Our analyses highlight that at the age where the majority of GABA-INS (Brief) have orientation preferences, preferences are well matched for both eyes and the observed difference does not significantly decrease with subsequent experience (Extended). In excitatory neurons, at the age where the majority of excitatory neurons display orientation preferences (Naive), significant differences in orientation preferences are observed for the eyes, and subsequent experience (Brief) does align orientation preferences. These differences highlight how the developmental trajectories of GABA-INS and excitatory neurons is not merely delayed, but also fundamentally different." The orientation preferences of the majority of GABA-INS are well matched for both eyes in Brief, and similar match is also observed for excitatory neurons in Brief. Then, the match in GABA-INS may just follow the match in excitatory neurons.

REVIEWER COMMENTS

Reviewer #1 (Remarks to the Author):

The authors have satisfactorily addressed my previous concerns.

We thank the reviewer for their feedback which has greatly improved our manuscript.

Reviewer #2 (Remarks to the Author):

In the new manuscript, the authors added additional data, most of which support the author's arguments and significantly improve the manuscript. However, I still have a few concerns. I recommend that these points should be addressed before publication.

1) In the new manuscript, the author performed a wavelet analysis to evaluate the modular activity patterns in the correlation map of the wide-field calcium imaging and reported the wavelength as a modularity metric. However, the wavelength indicates a distance between modules if the modularity exists, but does not support the existence of the modularity. Thus, the wavelength is not a good metric to confirm the modular patterns. Although the wavelength tells us the size of the modular organization if the wavelet fitting is good, some index for the existence of the modularity should be considered. When the modularity does not exist as the authors claim for Naïve, quantification of the wavelength would be meaningless,

We agree that the wavelength analysis alone does not indicate the presence of a modular network structure. To our knowledge, there has been no previously established metric for quantitatively assessing the existence of modular patterns. Therefore, we developed a new method for assessing the presence of modular patterns in our observations by comparing the experimentally measured wavelength to the wavelength computed for a flat correlation structure across our ROI. Wavelengths that are significantly different can be attributed to the existence of modular structure in the response correlation, while wavelengths that are not significantly different would suggest that modular structures are not present. Importantly, Naive animals did not have a significant difference in computed wavelength, suggesting that the measured wavelength could be attributed to a lack of modular response. The updated text has been included below (Lines 230-235).

Measured wavelengths for animals with brief experience were significantly smaller than uniform responses wavelengths (Brief 0.768 ± 0.033 mm vs Uniform 0.944 ± 0.035 mm, $n=4$, $p=0.0105$, Mean \pm SEM, Student's paired t-test), but they were not significantly different for Naive animals (Naive 0.890 ± 0.023 mm vs Uniform 0.893 ± 0.035 mm, $n=5$, $p=0.9270$, Mean \pm SEM, Student's paired t-test). Therefore, in Naive animals, measured wavelengths may reflect uniform correlation across the field of view and a lack of modular organization.

2) Related to 1), in the Supplementary fig 4c, the authors reported the comparable values of the

wavelength between excitatory and inhibitory cells in the BD and argue that large-scale modular organization in GABA-INs appears independent of visual experience (L353-354). However, in order to claim that the modular organization in GABA-INs is independent of visual experience, the wavelength in GABA-INs (and excitatory cells) should be compared with that in the normally reared, time-matching animals (i.e., Brief). Then, the wavelength of GABA-INs in BD appears higher than that in Brief and similar to that in Naïve (Supplementary fig. 3). This seems to suggest that the wavelength of modular pattern is sensory-dependent, which is inconsistent with the author's argument. The representative correlation map for BD shown in the supplementary fig 4b also seems less clear than the clear modular structure for Brief shown in fig 3c.

We appreciate that comparisons of brief experience and binocular deprivation would be informative. When we compare the wavelengths measured for GABA-INs in animals with brief experience and after binocular deprivation, we do not observe a significant difference in the wavelength. The appropriate statistics have been added to our manuscript and the additional text is reproduced below (lines 340-354).

These results indicate that experience plays a key role in the development of orientation-selective responses in GABA-INs, and one might expect that experience plays a similar role in the development of modular spatial organization for GABA-INs. First, we assessed whether large-scale organization of patterns was observable in BD and Recovery animals, using widefield epifluorescence imaging. Patterns of activity in both BD and Recovery animals displayed clustered, modular organization (Supplementary Fig. 4a,b). Consistent with our widefield observations, we observed diversity in the patterns for cellular trial activity (Supplementary Fig. 4d-f) in BD animals compared to Recovery animals. We further assessed the modularity of the pairwise pixel response correlations by computing the column spacing. Both BD and Recovery animals had comparable wavelengths to excitatory neurons (Supplementary Fig. 4c; BD excitatory 0.867 ± 0.065 mm vs GABA-INs 0.897 ± 0.064 mm, $p=0.6974$, $n=4$ vs 4; Recovery excitatory 0.796 ± 0.046 vs GABA-INs 0.798 ± 0.031 mm, $p=0.9618$, $n=5$ vs 4; Mean \pm SEM, bootstrap), and wavelengths for BD animals were comparable to GABA-INs in animals with normal visual experience (BD 0.897 ± 0.064 mm vs Brief 0.770 ± 0.053 mm, $p=0.1390$, $n=4$ vs 4; Mean \pm SEM, bootstrap). Thus, large-scale modular organization in GABA-INs appears independent of visual experience over the first week following normal eye opening.

3) Insufficient CcAMP expression resulted in the relatively small size of ROI (i.e., small size of robust visual evoked area, L592-593) in the wide-field imaging in the naïve, which does not completely discard the possibility that the modular pattern of visually evoked activity is missed in Naïve. The modular pattern of spontaneous activity in Naïve has recently been reported (Mulholland et al 2021). If the authors have observed the modular pattern of spontaneous activity during blank or baseline period in Naïve even within this small size of ROI, it would be really helpful to claim that the modular pattern cannot be missed even within this small size of ROI. This is optional, but descriptions about the spontaneous activity should ensure data quality and consistency with the published data, and also demonstrate a clear contrast between spontaneous and visually induced activity patterns which makes the author's argument more convincing. Alternatively, as commented in the author's reply, it

would be helpful if the authors address in the manuscript that the small size of the ROI does not fundamentally alters any of their analyses.

We thank the reviewer for this comment. We have presented two lines of evidence (the wavelength analysis in Supplementary Figure and the widefield presented in Fig 3e) that we believe together demonstrate that the modular patterns of GABA-INs in Naive animals are likely not present. In addition, we have a metric that does not depend on the size of the ROI (cellular correlation clustering Fig 3g) which again supports our previous findings. *Mulholland et al 2021* also explicitly tested the structure of activity for spontaneous- i.e. activity during long periods of no visual stimulation. Unfortunately, the relative lack of activity in our study during the brief blank trials precludes further analysis of spontaneous activity during our recordings.

Animals for which the area of viral expression was extremely limited (smaller than the expected wavelength for columnar organization), were excluded from our study. This means that we only used ROIs that were sufficiently large enough to observe modular structure. We show in Reviewer Fig. 1 (left) an example analysis of the wavelength for the example correlation pattern of an animal with Brief experience shown in Fig. 3b,c. Subsequently, we cropped the correlation map (right) using the smaller ROI for the Naive animals shown in Fig. 3b,c and recomputed the wavelength. Importantly, modular structure is clear even in the small ROI, and the wavelength is not drastically altered. Therefore, we still assert that the extent of expression associated with the size of our ROIs does not fundamentally alter our conclusions.

Reviewer Fig. 1 Small regions of interest are sufficiently large to capture modular structure. (A) Example full pairwise pixel correlation map for a Brief experience animal (left, reproduced from Fig. 2c). Cropped pairwise pixel correlation map for the Brief animal (right). The region of interest is the same size as the one for the Naive animal shown in Fig. 2c. Even with the smaller field of view a comparable wavelength was computed using the wavelet fitting column spacing method.

Furthermore, we ran a quantitative analysis to rule out the contribution of the size of the ROI in our computed column spacings. We calculated wavelength for flat correlation structure (as outlined above) in the ROIs for each age group and show that the wavelength is not significantly different. This demonstrates that for the range of wavelengths that we are testing for, the variation in size of the ROIs that we used does not impact the computed wavelength. We have updated our text to include this additional analysis and include an excerpt below (lines 225-229).

To control for uniform patterns of correlation, we computed wavelengths for uniform correlation patterns within our regions of interest. We found no significant difference in these wavelengths (Naive 0.894 ± 0.035 mm, $n=5$; Brief 0.944 ± 0.005 mm, $n=4$; Extended 0.908 ± 0.044 mm, $n=5$; Naive vs Brief $p=0.2507$; Naive vs Extended $p=0.8055$; Brief vs Extended $p=0.5032$; Mean \pm SEM, Student's t test), suggesting that the extent of expression did not impact our calculated wavelengths.

4) In Naïve animals, the visually evoked activity in GABA-INs shows wide-spread patterns, although the spontaneous activity is modular (Mulholland et al 2021). It would be helpful if the authors discuss what kind of circuit mechanisms can cause this difference.

As previously noted, we have acknowledged the observation of modular patterns of spontaneous activity by Mulholland et al. 2021 in our discussion. The observed differences are likely due to the primary source of excitatory drive for spontaneous activity versus visually evoked activity. As Smith et al. 2018 reported, most of the spontaneous activity observed in cortex is not generated in the retina or thalamus. Thus, spontaneous activity in the cortex likely strongly reflects intracortical connectivity. In contrast, visually-driven activity arises in the retina and the primary drive to cortex arises from the thalamus. This represents a fundamental difference in the origin of activity and suggests that our observed lack of organization of responses could arise from feedforward corticothalamic connections to GABA-INs. We have added in a reference to Smith et al. 2018 to further emphasize that spontaneous activity reflects intracortical connectivity and reproduced the appropriate paragraph below for your convenience (lines 446-462).

What contributes to the development of orientation preferences in GABA-INs? We observe that evoked responses show nearly uniform participation of interneurons in naive animals in primary visual cortex. Thus, it remains likely that a shared visual drive recruits the entire inhibitory network. One possibility is that uniform responses result from lateral inputs from neighboring layer II/III neurons that have a diverse spectrum of orientation preferences. In aggregate, these inputs could drive the non-orientation-specific responses seen early in development, and the subsequent refinement of the orientation preferences of these inputs could drive the gain of orientation tuning in interneurons. Alternatively, untuned responses in interneurons could arise from untuned inputs, possibly originating from the LGN or layer IV that override well-organized lateral inputs from layer II/III during visual stimulation, possibly masking modular GABA-IN network activity. The loss of these untuned inputs could unmask orientation tuned responses in interneurons after eye opening. Such a mechanism would be consistent with recent observations of spontaneous modular inhibitory activity before the onset of vision²¹- which likely reflect the structure of intracortical connections⁶- and anatomical studies that have established reduction in thalamocortical inputs to cortical interneurons across early development⁴⁶⁻⁴⁸. Further investigation into the synaptic inputs to GABA-INs is necessary to clarify these possibilities. In conclusion, our findings highlight that interneurons undergo a parallel, delayed developmental sequence in which experience-dependent processes drive the maturation of orientation-selective responses, resulting in modular, functional networks of GABA-INs.

5) The authors claim “Our analyses highlight that at the age where the majority of GABA-INs (Brief) have orientation preferences, preferences are well matched for both eyes and the observed difference does not significantly decrease with subsequent experience (Extended). In excitatory neurons, at the age where the majority of excitatory neurons display orientation preferences (Naive), significant differences in orientation preferences are observed for the eyes, and subsequent experience (Brief) does align orientation preferences. These differences highlight how the developmental trajectories of GABA-INs and excitatory neurons is not merely delayed, but also fundamentally different.” The orientation preferences of the majority of GABA-INs are well matched for both eyes in Brief, and similar match is also observed for excitatory neurons in Brief. Then, the match in GABA-INs may just follow the match in excitatory neurons.

We thank the reviewer for bringing this up and we realized that the way we had phrased this point might have been confusing. Our analyses highlight that the sequence of changes that defines the development of GABA-INs is not just a delayed version of the sequence displayed by excitatory neurons, but a fundamentally different sequence. Excitatory neurons exhibit an initial phase (Naive) where the majority display orientation preference, but with significant mismatch in orientation preferences for the two eyes. Subsequent experience (Brief) is necessary for binocular alignment of orientation preferences. In contrast, GABA-INs do not exhibit an initial phase with mismatched orientation preferences. Rather, when orientation selectivity first emerges for GABA-INs (Brief), the orientation preference they display is already well matched for both eyes, i.e., there is no period with mismatched orientation preference. Indeed, GABA-INs likely gain orientation-selective responses by receiving excitatory inputs from a network that has already achieved binocular aligned orientation-selective responses

Here is the new version that we think spells out what we intended and also includes that point the reviewer is making (lines 149-159).

Together, these results indicate that inhibitory and excitatory neurons have different developmental progressions for orientation selectivity and binocularly aligned responses. Excitatory neurons exhibit more robust orientation selectivity before the onset of experience, and this is accompanied by ocular differences in orientation preferences of binocular neurons. Subsequent visual experience drives binocular alignment of orientation preference and enhances the selectivity of excitatory neurons^{1,4}. In contrast, GABA-INs are only weakly orientation-selective before eye opening and increases in orientation selectivity arise concurrently with binocularly aligned preferences (without a misalignment phase) following the onset of visual experience. The nature of the synaptic changes that underlie the emergence of tuned responses in inhibitory neurons remains to be determined, but GABA-INs likely derive orientation-selective responses when the balance of their excitatory inputs arises from a network that has already achieved binocularly aligned responses.

REVIEWERS' COMMENTS

Reviewer #2 (Remarks to the Author):

The authors performed additional analyses and satisfied almost all my comments except one point. In the revised manuscript, the authors applied the wavelet analysis to uniform patterns and demonstrated that optimally fitted wavelengths were similar among Naïve, Brief, and Extended. Based on this, the authors argued that the size of the ROI did not affect the wavelength analysis. However, if a wavelet with a range of 0.5 to 1.0 mm wavelength (L619) was fitted to a flat pattern with a size beyond this range, resulting value will be close to the upper limit (i.e., ~1mm), and it should be natural to obtain the similar values among the three conditions. Thus, I think that this analysis is not appropriate and should be removed. However,

the authors suggested that the size of ROI in naïve is sufficient to observe the clustered activity by cropping a clustered pattern obtained in a Brief animal (Reviewer Fig. 1) and this would be a satisfactory control analysis. I recommend the authors to include this figure in the supplementary figures, rather than including the wavelet analysis.

Reviewer #2 (Remarks to the Author):

The authors performed additional analyses and satisfied almost all my comments except one point. In the revised manuscript, the authors applied the wavelet analysis to uniform patterns and demonstrated that optimally fitted wavelengths were similar among Naïve, Brief, and Extended. Based on this, the authors argued that the size of the ROI did not affect the wavelength analysis. However, if a wavelet with a range of 0.5 to 1.0 mm wavelength (L619) was fitted to a flat pattern with a size beyond this range, resulting value will be close to the upper limit (i.e., ~1mm), and it should be natural to obtain the similar values among the three conditions. Thus, I think that this analysis is not appropriate and should be removed. However, the authors suggested that the size of ROI in naïve is sufficient to observe the clustered activity by cropping a clustered pattern obtained in a Brief animal (Reviewer Fig. 1) and this would be a satisfactory control analysis. I recommend the authors to include this figure in the supplementary figures, rather than including the wavelet analysis.

We thank the reviewer for their feedback regarding our new analyses. We have split Supplementary Fig. 3 into two figures and have included the contents of Reviewer Fig. 1 demonstrating observable modular patterns of activity within a cropped ROI in our new Supplementary Fig. 3. We have also removed the analyses comparing flat correlations across ages. Below we have reproduced the updated text for reference (lines 209 –218):

*To further quantify the modularity of the correlation maps, we modified a method for measuring the column spacing (wavelength) of modular networks¹⁹. We observed comparable wavelengths for excitatory neurons and GABA-INS in experienced animals (Supplementary Fig. 3b; Brief excitatory 0.736 ± 0.032 mm vs GABA-INS 0.770 ± 0.053 mm, $n= 6$ vs 4 , $p= 0.5120$; Extended excitatory 0.797 ± 0.057 mm vs GABA-INS 0.844 ± 0.041 mm, $n= 5$ vs 5 , $p= 0.4730$; Mean \pm SEM, bootstrap test). In contrast, GABA-INS in Naive animals had a significantly larger wavelength (Naive excitatory 0.779 ± 0.020 vs GABA-INS 0.924 ± 0.026 , $n= 7$ vs 5 , $p= 0.0048$, Mean \pm SEM, bootstrap test), suggesting that responses are not as well associated with excitatory modular networks at eye opening. **Importantly, modular organization was still apparent and wavelengths were comparable when regions of interest for Naive animals were used to mask Brief experience correlation patterns (Supplementary Fig. 3c).***